# Restoration of locomotor function following stimulation of the A13 region in Parkinson's mouse models

**Linda H Kim[1,2†], Adam Lognon[1,2†], Sandeep Sharma[1,3], Michelle A Tran[3], Cecilia Badenhorst[1,3], Taylor Chomiak[1,4], Stephanie Tam[3], Claire McPherson[3], Todd E Stang[1,2], Shane EA Eaton[1,3], Zelma HT Kiss[1,2,4], Patrick J Whelan[1,3*]**

[1]Hotchkiss Brain Institute, University of Calgary, Calgary, Canada; [2]Department of Neuroscience, University of Calgary, Calgary, Canada; [3]Faculty of Veterinary Medicine, University of Calgary, Calgary, Canada; [4]Department of Clinical Neurosciences, University of Calgary, Calgary, Canada

**\*For correspondence:**
whelan@ucalgary.ca

[†]These authors contributed equally to this work

**Competing interest:** The authors declare that no competing interests exist.

## eLife Assessment

This **valuable** study reveals the pro-locomotor effects of activating a deep brain region containing diverse range of neurons in both healthy and Parkinson's disease mouse models. While the findings are **solid**, mechanistic insights remain limited due to the small sample size. This research is relevant to motor control researchers and offers clinical perspectives.

**Abstract** Parkinson's disease (PD) is characterized by extensive motor and non-motor dysfunction, including gait disturbance, which is difficult to treat effectively. This study explores the therapeutic potential of targeting the A13 region, a heterogeneous region of the medial zona incerta (mZI) containing dopaminergic, GABAergic, and glutamatergic neurons that has shown relative preservation in PD models. The A13 is identified to project to the mesencephalic locomotor region, with a subpopulation of cells displaying activity correlating to movement speed, suggesting its role in locomotion. We show that photoactivation of this A13 region can alleviate bradykinesia and akinetic features, while increasing turning in a mouse model of PD. These effects combine disease-specific rescue of function with a possible gain of function. We identified areas of preservation and plasticity within the A13 region using whole-brain imaging. Our findings suggest a global remodeling of afferent and efferent projections of the A13 region, highlighting the zona incerta's role as a crucial hub for the rapid selection of motor function. The study unveils the significant pro-locomotor effects of the A13 region and suggests its promising potential as a therapeutic target for PD-related gait dysfunction.

## Introduction

Parkinson's disease (PD) is a complex condition affecting many facets of motor and non-motor functions, including visual, olfactory, memory, and executive functions (**Cenci and Björklund, 2020**). Due to the widespread pathology of PD, focusing on changes within a single pathway cannot account for all symptoms. Gait dysfunction is one of the hardest to treat; pharmacological, deep brain stimulation (DBS), and physical therapies lead to only partial improvements (**Nonnekes et al., 2020**; **Nonnekes et al., 2015**). While the subthalamic nucleus (STN) and globus pallidus (GPi) are common DBS targets for PD, alternative targets such as pedunculopontine nucleus (PPN) and the zona incerta (ZI) have been proposed with mixed results in improving postural and/or gait

dysfunction (*Caire et al., 2013*; *Ferraye et al., 2010*; *Gut and Winn, 2015*; *Hamani et al., 2011*; *Moro et al., 2010*; *Nonnekes et al., 2015*; *Okun and Foote, 2010*; *Ossowska, 2020*; *Stefani et al., 2007*; *Thevathasan et al., 2018*). Most DBS work targeting the ZI has centered on areas close to the STN (*Ossowska, 2020*) and recent work shows responses linked to exploratory and goal-directed movements (*Hormigo et al., 2023*; *Sharma et al., 2024*). Recent work with photoactivation of subpopulations of PPN neurons in PD models shows promise for similar ZI-focused strategies (*Masini and Kiehn, 2022*). Indeed, our recent work on preclinical models shows that DBS of the A13 in rat models effectively produces locomotor activity that can be incorporated into ongoing behavior (*Bisht et al., 2025*).

The ZI is recognized as an integrative hub, with roles in regulating sensory inflow, arousal, motor function, and conveying motivational states (*Chometton et al., 2017*; *Mitrofanis, 2005*; *Monosov et al., 2022*; *Sharma et al., 2024*; *Wang et al., 2020*; *Yang et al., 2022*; *Zhao et al., 2019*). As such, it is well placed to be involved in PD and has seen increased clinical and preclinical research over the last two decades (*Blomstedt et al., 2018*; *Ossowska, 2020*; *Plaha et al., 2008*). However, little attention has been paid to the medial zona incerta (mZI), particularly the A13, the only dopamine-containing region of the rostromedial ZI (*Bolton et al., 2015*; *Kim et al., 2017*; *Sharma et al., 2018*). Recent research in primates and mice (*Peoples et al., 2012*; *Roostalu et al., 2019*; *Shaw et al., 2010*) indicates that the A13 is preserved in 1-methyl-4-phenyl-1,2,3,6-tetrahydropyridine (MPTP)-based PD models. Yet it is not clear whether the A13 region substantially remodels in PD animal models as has been observed for other areas of the brain (*Ji et al., 2023*).

Recently, we discovered that the A13 located within the ZI projects to two areas of the mesencephalic locomotor region (MLR), the PPN and the cuneiform nucleus (CUN) (*Sharma et al., 2018*), suggesting a role for A13 in locomotor function. Indeed, mini-endoscopic calcium recordings from calcium/calmodulin-dependent protein kinase IIα (CaMKIIα) populations in the rostral ZI, which includes the A13 nucleus, show a subpopulation of cells whose activity correlates with movement speed (*Li et al., 2021*). As this region projects to the MLR, it is a potential motor pathway to target for gait improvement, which has been substantiated by our DBS work targeting the A13 in rats (*Bisht et al., 2025*). Photoactivation of glutamatergic MLR neurons alleviates motor deficits in mouse models that either transiently blocked dopamine transmission or lesioned substantia nigra pars compacta (SNc) with 6-hydroxydopamine (6-OHDA) (*Fougère et al., 2021*; *Masini and Kiehn, 2022*). Phenomena such as kinesia paradoxa (*Glickstein and Stein, 1991*) in PD patients support the existence of preserved parallel motor pathways that can be engaged in particular circumstances to produce normal movement.

Additional support for parallel motor pathways in PD comes from studies showing functional changes in A13 (*Hoffman et al., 1997*; *Périer et al., 2000*). Nigrostriatal lesions affect cellular function and lead to anatomical remodeling in monoaminergic brain regions including widespread alterations in dopaminergic, noradrenergic, cholinergic, and serotoninergic neuronal populations; however, global connectivity patterns from A13 have not been explored (*Braak et al., 2003*; *Kish et al., 2008*; *Lim et al., 2009*; *Perez-Lloret and Barrantes, 2016*; *Roostalu et al., 2019*; *Scatton et al., 1983*; *Zweig et al., 1989*). There is additional evidence showing parallel motor pathways in the A13. For example, the A13 connectome encompasses the cerebral cortex (*Mitrofanis and Mikuletic, 1999*), central nucleus of the amygdala (*Eaton et al., 1994*), thalamic paraventricular nucleus (*Li et al., 2014*), thalamic reuniens (*Sita et al., 2007*; *Venkataraman et al., 2021*), CUN and PPN (*Sharma et al., 2018*), superior colliculus (SC) (*Bolton et al., 2015*), and dorsolateral periaqueductal gray (PAG) (*Messanvi et al., 2013*; *Sita et al., 2007*), making the A13 a potential hub for goal-directed locomotion (*Choi and McNally, 2017*; *Eaton et al., 1994*; *Messanvi et al., 2013*; *Mok and Mogenson, 1986*; *Moriya et al., 2020*; *Ogundele et al., 2017*; *Sanghera et al., 1991a*; *Sanghera et al., 1991b*; *Venkataraman et al., 2021*).

Given the A13 region's role in gait control and its therapeutic potential in PD, we investigated the effects of its photoactivation in 6-OHDA mouse models. The targeted area includes the A13 and a portion of the mZI, collectively referred to as the A13 region throughout this study. Photoactivation of the A13 region alleviated bradykinetic and akinetic symptoms in a mouse model of unilateral nigrostriatal degeneration induced by 6-OHDA. Our exploratory work on remodeling of input and output patterns in the A13 region in 6-OHDA mice suggests potential downstream targets mediating the effects of photoactivation. These findings demonstrate that the A13 region exerts

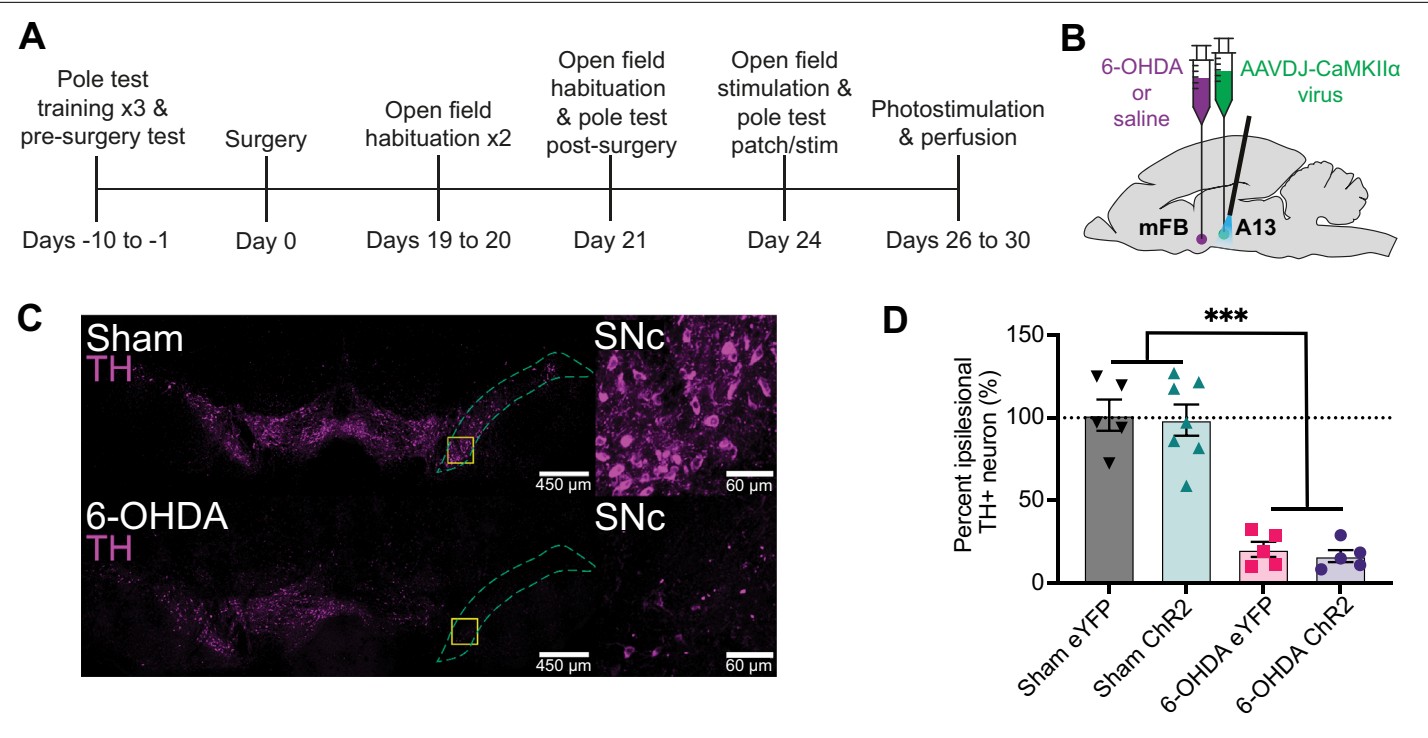

**Figure 1.** Experimental design and confirmation of unilateral TH⁺ depletion in the SNc via 6-hydroxydopamine (6-OHDA) lesion. (**A**) Illustration of experimental timeline. (**B**) Dual ipsilateral stereotaxic injection into the medial forebrain bundle (mFB) and A13 region. (**C**) TH⁺ cells in the SNc of a sham animal (top) compared to a 6-OHDA-injected mouse (bottom). Magnified areas, outlined by yellow squares, are shown at right. (**D**) Unilateral injection of 6-OHDA into the mFB (6-OHDA ChR2: $n = 5$; 6-OHDA eYFP: $n = 5$) resulted in a significantly greater percentage loss of TH⁺ cells in the SNc compared to sham animals (sham ChR2: $n = 7$ ; sham eYFP: $n = 5$), regardless of virus type (two-way ANOVA: $F_{1,18} = 104.4$, $p < 0.001$). ***$p < 0.001$. Error bars indicate SEMs.

The online version of this article includes the following source data for figure 1:

**Source data 1.** Raw data and statistical results of the percentage loss of TH⁺ cells in the SNc in 6-hydroxydopamine (6-OHDA) and sham animals.

strong pro-locomotor effects in both normal and PD mouse models. Portions of these data have been presented previously in abstract form (*Kim et al., 2021*).

## Results

### Unilateral 6-OHDA mouse model has robust motor deficits

The overall experimental design is illustrated in *Figure 1A*, along with a schematic in *Figure 1B* showing injections of 6-OHDA in the medial forebrain bundle (mFB) and AAVDJ-CaMKIIα-ChR2 virus into the mZI. We confirmed SNc degeneration in a well-validated unilateral 6-OHDA-mediated Parkinsonian mouse model (*Thiele et al., 2012*). The percentage of tyrosine hydroxylase (TH⁺) cell loss normalized to the intra-animal contralesional side was quantified. 6-OHDA produced a significant lesion that decreased TH⁺ neuronal SNc populations. As previously reported (*Boix et al., 2015*), the SNc ipsilesional to the 6-OHDA injection ($n = 10$) showed major ablation of the TH⁺ neurons compared to sham animals (*Figure 1C, D*: $n = 12$).

### A13 region photoactivation generates pro-locomotor behaviors in the open field

6-OHDA lesions produce bradykinetic and akinetic phenotypes in the open field (*Li et al., 2022*; *Magno et al., 2019*; *Sanders and Jaeger, 2016*). We first confirmed localization of the optical fiber above the A13 region, centered on the mZI, along with YFP reporter expression in mice given sham or 6-OHDA injections (*Figure 2*; *Figure 2—figure supplement 1*). Corroborating the post hoc targeting, we found evidence for c-Fos in neurons within the A13 region in photostimulated ChR2

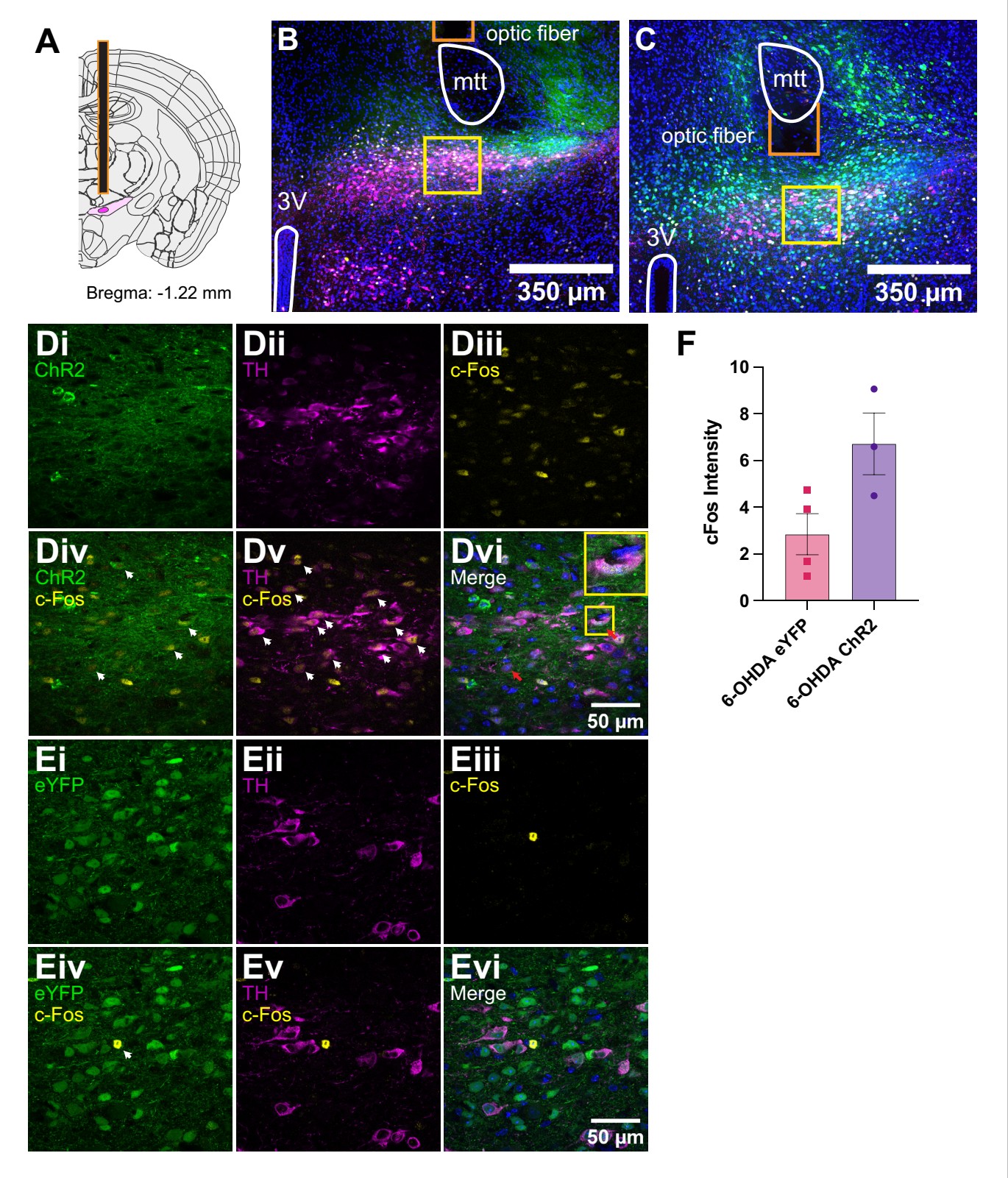

**Figure 2.** Post hoc c-Fos expression and targeting of the mZI and A13. (**A**) Diagram showing the A13 dopaminergic (DAergic) nucleus in dark magenta, encapsulated by the zona incerta (ZI) in light magenta. The fiber-optic tip is outlined in red. Atlas image adapted from the Scalable Brain Atlas (**Lein et al., 2007**; **Bakker et al., 2015**). (**B**) Tissue images were obtained from a 6-hydroxydopamine (6-OHDA) ChR2 animal and (**C**) a 6-OHDA eYFP animal. Images show the distribution of DAPI (blue), eYFP (green), c-Fos (yellow), and TH (magenta). Landmarks are outlined in white (3V: third ventricle; A13 and

*Figure 2 continued on next page*

*Figure 2 continued*

mZI as shown in A), and the optic cannula tip is shown in red. Higher-magnification images of the A13 region are outlined by yellow boxes in a 6-OHDA ChR2 animal (**Di–Dvi**) and a 6-OHDA eYFP animal (**Ei–Evi**). Images show isolated channels in the top rows of each group: (**i**) eYFP, (**ii**) TH, and (**iii**) c-Fos. Merged channels are shown in the bottom rows: (**iv**) eYFP/ChR2 + c-Fos, (**v**) TH + c-Fos, (**vi**) all three channels merged. White arrowheads in the merged images highlight areas of marker overlap. Red arrows indicate triple colocalization of ChR2, c-Fos, and TH. (**Dvi**) contains a magnified example of triple-labeled neurons, highlighted in the yellow box. (**F**) Graph shows an increase in c-Fos fluorescence intensity after photoactivation in 6-OHDA ChR2 mice ($p = 0.05$).

The online version of this article includes the following source data and figure supplement(s) for figure 2:

**Source data 1.** Raw data and statistical results of cFos intensity after photoactivation in 6-hydroxydopamine (6-OHDA) ChR2 animals.

**Figure supplement 1.** Quantification of channelrhodopsin viral spread in the rostral–caudal direction from the injection site in 6-hydroxydopamine (6-OHDA)-treated and sham animals.

**Figure supplement 1—source data 1.** Processed histogram and raw data from regions of interest (ROIs) using ImageJ to calculate the percentage area spread of CHR2 virus in sham and 6-hydroxydopamine (6-OHDA) animals.

mice (*Figure 2D–F*). Before post hoc analysis, mice were monitored in the open-field test (OFT), where the effects of the 6-OHDA lesion were apparent, with 6-OHDA lesioned animals demonstrating less movement, fewer bouts of locomotion, and less time engaging in locomotion (*Figure 3A–I*). Using instantaneous animal movement speeds that exceeded 2 cm/s as per *Masini and Kiehn, 2022*, we plotted instantaneous speed (*Figure 3B–E*) and analyzed 1-min bins (*Figure 3H*). As was expected, 6-OHDA lesioned animals had lower movement speeds than sham control animals ($p < 0.001$). One animal from the 6-OHDA eYFP group was excluded because it did not meet the speed threshold during recording. Notably, photoactivation of the A13 region often generated dramatic effects, with mice showing a distinct increase in locomotor behavior (*Figure 3A*, *Figure 3—videos 1 and 2*). Both sham and 6-OHDA ChR2 mice showed a significant increase in locomotor distance traveled during periods of photoactivation (*Figure 3F*, $p = 0.005$). One sham animal showed grooming behavior on stimulation and was excluded from the analysis.

We tested whether photoactivation led to a single bout of locomotion or if there was an overall increase in bouts, signifying that animals could repeatedly initiate locomotion following photoactivation. Mice in the ChR2 groups demonstrated an increase in the number of locomotor bouts with photoactivation, indicating a greater ability to start locomotion from rest, and that photoactivation was not eliciting a single prolonged bout (*Figure 3G*, $p = 0.005$). Photoactivation also increased the total duration of locomotor bouts (*Figure 3I*, $p < 0.001$). We did note a refractory decrease in the distance traveled by the sham ChR2 group. To control for this, we compared the pre-stimulation time points to the baseline 1-min averages to ensure that the animal locomotion distance traveled returned to a stable state before stimulation was reapplied (*Figure 3—figure supplement 1*, $p = 0.78$).

Next, we examined the reliability of photoactivation to initiate locomotion. The percentage of trials with at least one bout of locomotion was compared for the pre- and stim time points in 6-OHDA mice. 6-OHDA ChR2 animals showed a reliable pro-locomotion phenotype with A13 region photoactivation (*Figure 3—figure supplement 2A*, $p = 0.042$). As was expected in the control 6-OHDA eYFP group, there was no effect of photoactivation on the probability of engaging in locomotion (*Figure 3—figure supplement 2A*, $p = 0.71$).

Movement speed contributes to total distance traveled and reflects the bradykinetic phenotype observed in 6-OHDA-lesioned mice (*Magno et al., 2019*; *Sanders and Jaeger, 2016*). The 6-OHDA and sham ChR2 groups displayed increases in average speed in comparison to the 6-OHDA and sham eYFP groups during photoactivation (*Figure 3H*, $p < 0.001$). There was no difference in the time to initiate locomotion between the sham and 6-OHDA ChR2 groups (*Figure 3—figure supplement 2B*, $p = 0.95$).

## Photoactivation of the A13 region increases ipsilesional turning in the OFT

Unilateral 6-OHDA lesions drive asymmetric rotational bias (*Boix et al., 2015*; *Li et al., 2022*; *Magno et al., 2019*; *Thiele et al., 2012*). We investigated whether rotational bias persisted during photoactivation and observed that 6-OHDA ChR2 animals displayed increased ipsilesional rotation. Specifically, these animals showed a significant increase in turn angle sum (TAS), indicating enhanced rotational

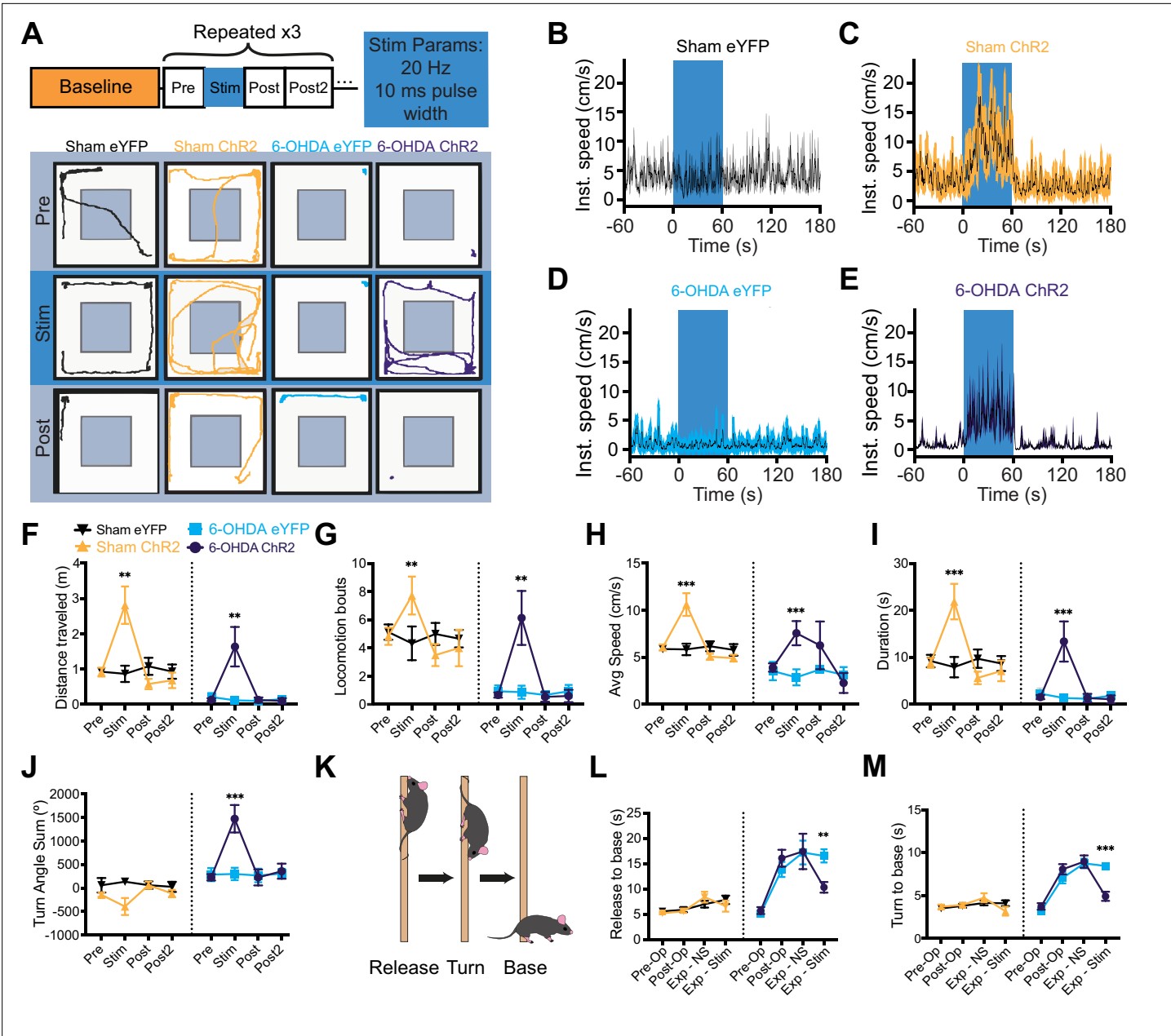

**Figure 3.** Ipsilesional photoactivation of the A13 region in a unilateral 6-hydroxydopamine (6-OHDA) mouse model rescues motor deficits.
(**A**) Schematic of the open-field experiment design and example traces from open-field testing. Each testing bin represents 1 min (total duration: 4 min) with unilateral photoactivation of the A13 region. There were three technical replicates performed per mouse. Group-averaged instantaneous velocity graphs showing no increase in a sham eYFP (**B**) or 6-OHDA eYFP mouse (**C**), and increased velocity during stimulation in a sham ChR2 (**D**) and 6-OHDA ChR2 (**E**) mouse. (**F–I**) Effects of photoactivation on open-field metrics for sham eYFP (n = 5), sham ChR2 (n = 6), 6-OHDA eYFP (n = 5), and 6-OHDA ChR2 (n = 5) groups. Statistical comparisons used three-way mixed-model ANOVAs with post hoc Bonferroni pairwise tests. Photoactivation increased locomotor activity in both sham and 6-OHDA ChR2 groups for the following: (**F**) distance traveled (ChR2 vs. eYFP: p = 0.005), (**G**) locomotor bouts (ChR2 vs. eYFP: p = 0.005), (**H**) movement speed (ChR2 vs. eYFP: p < 0.001), and (**I**) duration of locomotion in the open field (ChR2 vs. eYFP: p < 0.001). (**J**) The graph presents animal rotational bias using the turn angle sum. A significant increase in rotational bias was observed in 6-OHDA ChR2 mice during A13 region photoactivation (6-OHDA ChR2 vs. 6-OHDA eYFP: p < 0.001). (**K**) Diagram of the pole test. A mouse is placed facing upward on a vertical pole; 'time to release' is defined as the interval from the experimenter removing their hand from the animal's tail to when the animal touches the ground. (**L, M**) Photoactivation of the A13 region decreased the time required to navigate to the base in 6-OHDA ChR2 mice compared to 6-OHDA eYFP mice (p = 0.004). A pre-op baseline was performed, followed by post-op testing 3 weeks later. On the experiment day, performance with no stimulation (Exp – NS) was compared to photoactivation (Exp – Stim). (**M**) 6-OHDA ChR2 mice showed a further reduction in time to reach the base compared to 6-OHDA eYFP mice (6-OHDA ChR2 vs. 6-OHDA eYFP: p < 0.001). ***p < 0.001, **p < 0.01. Error bars indicate SEMs.

*Figure 3 continued on next page*

*Figure 3 continued*

The online version of this article includes the following video, source data, and figure supplement(s) for figure 3:

**Source data 1.** Raw data and statistical results of distance traveled in 6-hydroxydopamine (6-OHDA) and sham animals.

**Figure supplement 1.** Time course of open-field locomotion distance traveled over 30 min.

**Figure supplement 1—source data 1.** Normalized distance traveled data for sham ChR2 animals at baseline and across five pre-stimulation time points.

**Figure supplement 1—source data 2.** Characterization of A13 region photoactivation temporal dynamics on locomotion initiation.

**Figure supplement 2.** Characterization of A13 region photoactivation temporal dynamics on locomotion initiation.

**Figure 3—video 1.** Photoactivation of the A13 region in a 6-hydroxydopamine (6-OHDA) model mouse producing increased locomotion in the open-field test (OFT) (2x speed).

https://elifesciences.org/articles/90832/figures#fig3video1

**Figure 3—video 2.** Photoactivation of the A13 region in a sham mouse producing increased locomotion in the open-field test (OFT) (2x speed).

https://elifesciences.org/articles/90832/figures#fig3video2

**Figure 3—video 3.** Photoactivation of the A13 region during the pole test in a 6-hydroxydopamine (6-OHDA) model mouse decreases pole descent time (0.5x speed).

https://elifesciences.org/articles/90832/figures#fig3video3

bias with photoactivation toward the lesioned side (*Figure 3J*, $p < 0.001$). As expected, 6-OHDA eYFP animals maintained a consistent rotational bias over time. To determine whether this effect was due to photoactivation alone or its interaction with the lesion, we also analyzed sham ChR2 animals. This group showed no significant change in TAS with photoactivation (*Figure 3J*, $p = 0.06$), suggesting that the effect is lesion-dependent. Next, we assessed whether the elevated TAS in 6-OHDA ChR2 animals occurred during locomotion. When TAS was calculated only during locomotor periods, the rotational bias was no longer significant ($p > 0.05$).

## Skilled vertical locomotion is improved in the pole test with photoactivation of the A13 region

The pole test is a well-established behavioral assay for 6-OHDA models (*Figure 3K*), requiring skilled locomotion for the animal to turn and descend a vertical pole (*Matsuura et al., 1997*; *Ogawa et al., 1985*). Improvements in function can be inferred if the time taken to complete the test decreases (*Matsuura et al., 1997*; *Ogawa et al., 1985*). 6-OHDA mice demonstrated significantly greater times navigating to the base than sham mice (*Figure 3L*, $p = 0.004$). Photoactivation of the A13 region led to shorter descent times to the base of the pole in 6-OHDA ChR2 mice compared to 6-OHDA eYFP mice (*Figure 3L*, $p = 0.004$, *Figure 3—video 3*). Neither of the eYFP groups showed any changes in the time to complete the pole test (*Figure 3L*).

To isolate the effects of photoactivation on descent ability, we analyzed the time taken to descend after the turn, excluding delays from exploratory behavior at the top of the pole. While all groups showed reduced total pole test descent time with photoactivation, considering just the time to descend from turn alone, there was a larger improvement with A13 region photoactivation in the 6-OHDA ChR2 mice compared to 6-OHDA eYFP mice (*Figure 3M*, $p < 0.001$). These results indicate that photoactivation has the effect of reducing bradykinesia by improving the ability of mice to descend the pole during the pole test.

## Dopaminergic cells in the A13 region are preserved in the unilateral 6-OHDA mouse model

While photoactivation of the A13 region increased locomotor activity in both sham and 6-OHDA lesioned mice, we observed differences in speed and directional bias between the two groups. We hypothesized that these differences might be due to changes in the A13 region's connectome since previous research has shown that 6-OHDA lesions can lead to increases in firing frequency and metabolic activity in this brain region (*Périer et al., 2000*). To investigate this possibility, we used whole-brain imaging approaches (*Hansen et al., 2020*; *Zhan et al., 2021*) to examine changes in the connectome following 6-OHDA lesions of the nigrostriatal pathway.

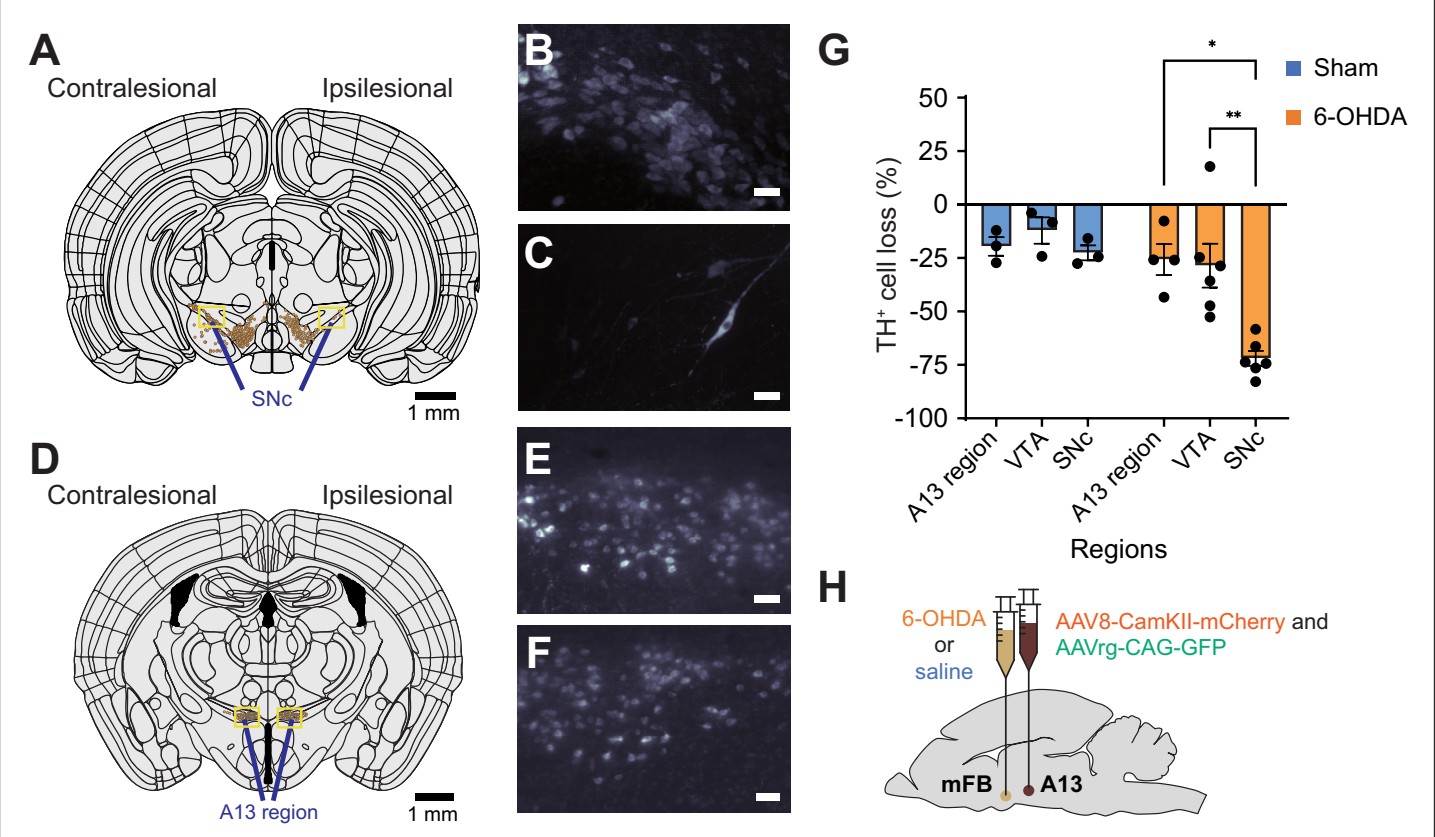

**Figure 4.** Preservation of TH+ A13 cells in Parkinsonian mouse models. Representative slices of SNc (AP: –3.08 mm, **A**) and A13 region (AP: –1.355 mm, **D**) following registration with WholeBrain software.There was a loss of TH+ SNc cells following 6-hydroxydopamine (6-OHDA) injections at the medial forebrain bundle (mFB) (**A**). (**B, C**) Zoomed sections (90 µm thickness) of red boxes in panel A in left to right order. Meanwhile, TH+ VTA cells were preserved bilaterally. Additionally, TH+ A13 cells were present on the ipsilesional side to 6-OHDA injections (**D**). (**E, F**) Zoomed sections (90 µm thickness) of red boxes in panel D in left to right order. When calculating the percentage of TH+ cell loss normalized to the intact side, there was a significant interaction between the condition and brain region (repeated measures two-way ANOVA with post hoc Bonferroni pairwise, sham: n = 3, 6-OHDA: n = 6). (**G**) 6-OHDA-treated mice showed a significantly greater percentage of TH+ cell loss in SNc compared to the VTA and A13 region (VTA vs. SNc: p = 0.004; A13 region vs. SNc: p = 0.012). In contrast, sham showed no significant difference in TH+ cell loss across SNc, VTA, and A13 regions (p > 0.05). *p < 0.05 and **p < 0.01. (**H**) Dual ipsilateral stereotaxic injection into the mFB and A13 region. Scale bars are 50 µm unless otherwise indicated.

The online version of this article includes the following source data and figure supplement(s) for figure 4:

**Source data 1.** Raw data and statistical results of TH+ cell loss in 6-hydroxydopamine (6-OHDA) and sham animals.

**Figure supplement 1.** Injection core in a sham brain showing viral tracer spread in the A13 region.

As expected (*Iancu et al., 2005*), our whole-brain imaging results showed that TH+ cells in SNc (*Figure 4B, C*) were more vulnerable to the 6-OHDA neurotoxin than those in the A13 (*Figure 4E, F*). Specifically, we found that 6-OHDA-treated mice showed a significantly greater percentage of TH+ cell loss in SNc compared to the VTA and A13 (*Figure 4G*; VTA vs. SNc: p < 0.01; A13 vs. SNc: p < 0.01). In contrast, sham animals showed no change in TH+ cell numbers across SNc, VTA, and A13 (*Figure 4G*, p > 0.05). These findings are consistent with observations in the human brain, where the A13 region is preserved in the presence of extensive nigrostriatal degeneration (*Matzuk and Saper, 1985*). Our results confirm that the 6-OHDA mouse model effectively replicates this aspect of PD pathology.

## Extensive remodeling of the A13 region connectome following unilateral nigrostriatal degeneration

Although photoactivation of the A13 region was effective in restoring speed in 6-OHDA lesioned mice (*Figure 3H*), we observed an increase in circling behavior (*Figure 3J*). This suggested that additional changes possibly reflecting alterations in the A13 connectome may be occurring. To investigate these potential changes without the potential confounds of an implanted optrode over the area, we

conducted separate experiments to examine the changes in the input and output patterns of the A13 region in a small cohort of sham and 6-OHDA mice using whole-brain imaging approaches. We did this by co-injecting anterograde (AAV8-CamKIIα-mCherry) and retrograde AAV (AAVrg-CAG-GFP) tracers into the A13 nucleus (*Paxinos and Franklin, 2008*; *Figure 4H*). The injection core and spread were determined in the rostrocaudal direction from the injection site (*Figure 4—figure supplement 1*). The viral spread was centered around the mZI containing the A13, with minor spread to adjacent areas in some cases (*Figure 4—figure supplement 1*). To assess whether unilateral nigrostriatal degeneration led to changes in the organization of motor-related inputs and outputs from the A13, we first visualized interregional correlations of afferent and efferent proportions for each condition using correlation matrices (*Figure 5A, B*; 18 regions in a pairwise manner). The goal of this analysis was not to infer mechanistic pathways, but rather to provide a systems-level overview of how the global organization of A13 efferents and afferents is altered following 6-OHDA lesioning, highlighting how groups of regions co-vary in their input to or output from the A13 region. For example, a positive correlation between inputs from Region A and Region B to the A13 suggests that across animals, when input from Region A is relatively high, input from Region B tends to be high as well, indicating that connectivity from these regions to the A13 may be co-regulated or affected similarly by the lesion. Conversely, a shift from positive to negative correlation may signal a divergence in how regions contribute to the A13 connectome after nigrostriatal degeneration. Correlation matrices were organized using the hierarchical anatomical groups from the Allen Brain Atlas (*Figure 5C–F*). To control for experimental variations in the total labeling of neurons and fibers, we calculated the proportion of total inputs and outputs by dividing the afferent cell counts or efferent fiber areas in each brain region by the total number found in the brain. The data were then normalized to a $\log_{10}$ value to reduce variability and bring brain regions with high and low proportions of cells and fibers to a similar scale (*Kimbrough et al., 2020*). Comparing the afferent and efferent proportions pairwise between mice showed good consistency with an average correlation of 0.91 ± 0.02 (Spearman's correlation, *Figure 6—figure supplement 1*).

We observed differences in afferent input patterns to the A13 region between sham and 6-OHDA groups. Correlation matrix analysis revealed that motor-related inputs to the A13 in sham animals exhibited stronger positive correlations across cortical, subcortical, and brainstem regions compared to 6-OHDA mice (*Figure 5C*). In 6-OHDA-lesioned mice, input distribution from dorsal pallidum, lateral hypothalamus, ZI, and tegmental reticular nucleus became positively correlated to somatomotor (MOp, MOs) and somatosensory (SSp) cortical areas (*Figure 5D*). Notably, these regions were anti-correlated with other motor-related inputs, indicating a reorganization of the afferent network. These data suggest a shift in the A13 inputs after 6-OHDA lesions, with a relative increase in cortical, pallidal, and hypothalamic influence compared to other motor-related brain regions.

In contrast, output patterns from the A13 region showed a higher overall correlation between brain regions in 6-OHDA-lesioned mice. In sham animals, A13 outputs to cortical regions were negatively correlated with outputs to thalamic, hypothalamic, and midbrain regions, indicating a structured and selective projection pattern (*Figure 5E*). This specificity was lost (*Figure 5F*) in 6-OHDA mice, broader, less targeted distribution of A13 outputs following dopaminergic degeneration. These patterns offer new insight into the broader reorganization of the A13 connectome and may serve as systems-level signatures of altered anatomical organization, providing a foundation for future mechanistic investigations using circuit-specific tools. Future studies using cell type- and/or projection-specific functional manipulations will be essential to determine the causal roles of these reorganized circuits.

## Differential remodeling of the A13 region connectome ipsi- and contra-lesion following 6-OHDA-mediated nigrostriatal degeneration

We used whole-brain imaging to further investigate how the A13 connectome is affected by a unilateral 6-OHDA lesion. Our analysis revealed distinct patterns of remodeling on the lesioned (ipsilesional) and intact (contralesional) sides of the brain (*Figure 6B, C, E, F*). On the ipsilesional side, we observed a decrease in A13 afferent density in 6-OHDA mice compared to sham animals from several key regions (*Figure 6A*), including the primary motor cortex (MOp), primary somatosensory area (SSp), and secondary motor cortex (MOs). This reduced input from MOp, which is known for its role in initiating movement, could contribute to the initial bradykinesia observed in 6-OHDA mice. In contrast, ipsilesional compensatory increases in A13 afferents were observed, for example, from the

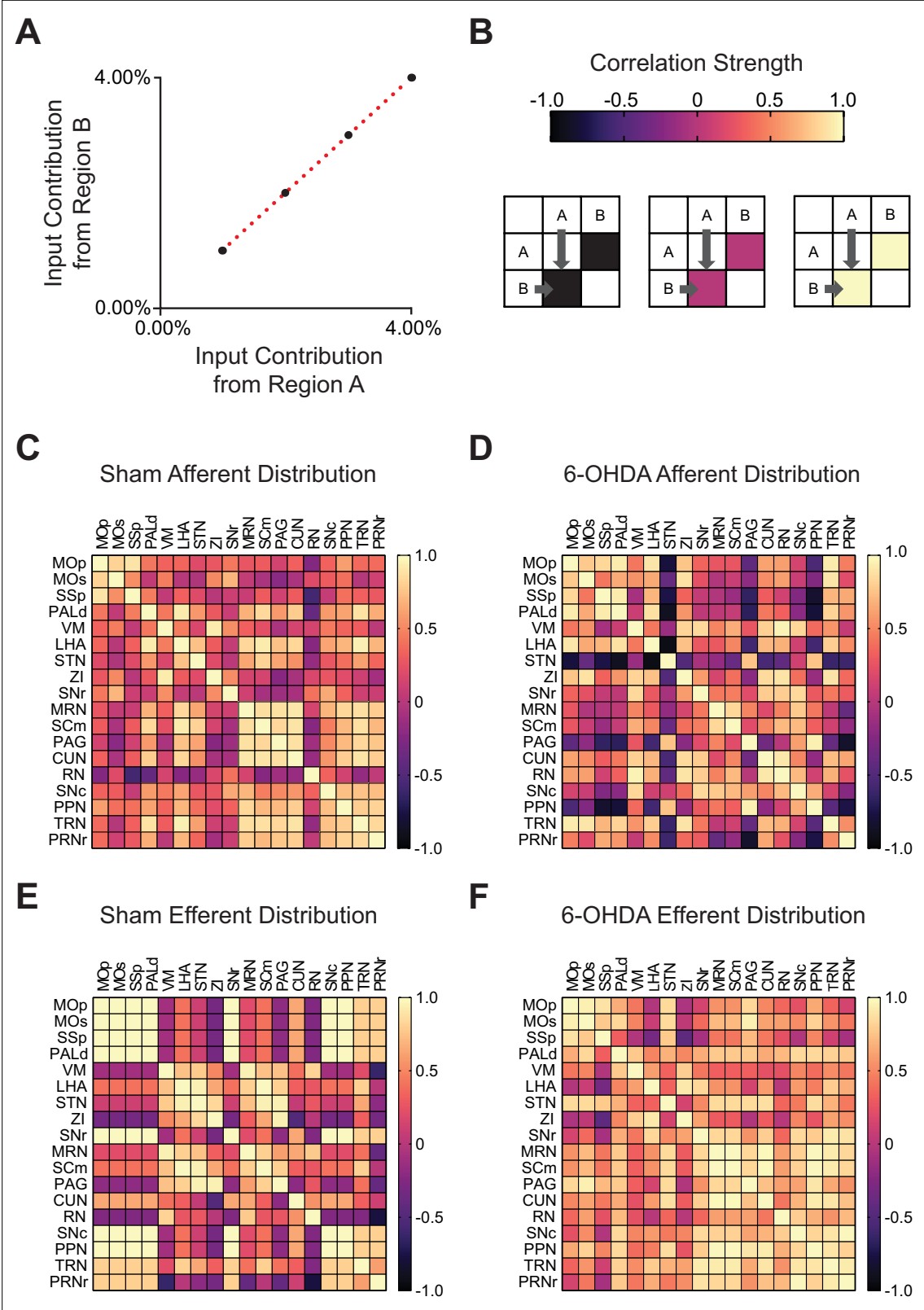

**Figure 5.** Nigrostriatal degeneration causes widespread changes in A13 region input and output connections. Correlation matrices were used to visualize the input and output patterns of the A13 region, focusing on motor-related pathways. (**A**) Brain regions with similar input patterns exhibit strong correlations. (**B**) Correlation strength is represented by cell color in the matrix: yellow indicates strong positive correlations, magenta denotes no correlation, and black indicates strong negative correlations. (**C, D**) Sham animals displayed stronger interregional correlations among inputs from

*Figure 5 continued on next page*

*Figure 5 continued*

motor-related regions across the neuraxis to the A13 region compared to 6-hydroxydopamine (6-OHDA)-lesioned mice. This suggests a broader distribution of inputs among motor-related cortical, subcortical, and brainstem regions in sham animals. (**D**) In 6-OHDA lesioned mice, inputs to the A13 region from the STN, PAG, and PPN became negatively correlated, unlike inputs from other motor-related regions. In contrast, inputs from motor-related pallidal and incertohypothalamic areas showed stronger positive correlations with cortical inputs, suggesting these regions may exert greater influence on A13 activity. (**E, F**) In contrast, output patterns from the A13 region showed stronger interregional correlations among cortical and brainstem motor-related regions in 6-OHDA-lesioned mice compared to sham animals. (**E**) In sham animals, A13 outputs to cortical regions were negatively correlated with outputs to thalamic, hypothalamic, and midbrain regions. This pattern was lost following nigrostriatal degeneration, suggesting a more distributed pattern of A13 outputs. MOp (primary motor cortex), MOs (secondary motor cortex), SSp (primary somatosensory area), PALd (pallidum, dorsal), VM (ventral medial thalamic nucleus), LHA (lateral hypothalamus), STN (subthalamic nucleus), ZI (zona incerta), SNr (substantia nigra pars reticulata), MRN (midbrain reticular nucleus), SCm (superior colliculus, motor), PAG (periaqueductal gray), CUN (cuneiform nucleus), RN (red nucleus), SNc (substantia nigra pars compacta), PPN (pedunculopontine nucleus), TRN (tegmental reticular nucleus), and PRNr (pontine reticular nucleus).

The online version of this article includes the following source data and figure supplement(s) for figure 5:

**Source data 1.** Cross-correlation data across A13 region inputs and outputs in 6-hydroxydopamine (6-OHDA) and sham animals.

**Figure supplement 1.** Ipsilateral (**A–F**) and contralateral (**G–L**) afferent and efferent proportions in sham (blue) and 6-hydroxydopamine (6-OHDA) (orange) mice.

**Figure supplement 1—source data 1.** Dataset of A13 counts or pixels.

lateral hypothalamic area (LHA), substantia nigra pars reticulata (SNr), superior colliculus (SCm), and PAG (*Figure 6A*). Examples are shown in *Figure 6—figure supplement 1*.

Interestingly, the contralesional side showed a more conservative pattern of afferent remodeling, with a modest decrease in A13 afferent density in primary motor cortex (MOp) and primary somatosensory cortex (SSp) of 6-OHDA mice, with a slight increase in the secondary motor cortex (MOs) (*Figure 6A*). For hypothalamic and midbrain structures, only LHA and PAG showed evidence for an increase. Several regions, including the hypothalamus, midbrain, and pons, showed bilateral upregulation of A13 afferents, suggesting a more global compensatory response to the unilateral lesion.

Ipsilesional A13 efferents were decreased in 6-OHDA mice (*Figure 6F*) mainly in the somatosensory cortex (SSp; *Figure 6D*, *Figure 6—figure supplement 2*), with some modest increases in midbrain structures, whereas contralesional efferent projection patterns showed increases in efferent density in 6-OHDA mice compared to sham in cortical structures (MOp, MOs, and SSp). This increase in efferent projections to the contralesional motor cortices could explain the increased ipsilesional turning bias we observed during A13 photoactivation in the 6-OHDA group (*Figure 3J*). This was accompanied by modest decreases in efferent density in 6-OHDA contralesional midbrain structures compared to sham. Examples are shown in *Figure 6—figure supplement 2*.

## Discussion

This study aimed to determine whether activation of the A13 region could influence locomotor function, particularly in a PD model. Our results show that photoactivation of the A13 region enhances locomotion in both lesioned and sham mice, increasing distance traveled, locomotion time, and speed. Particularly in 6-OHDA mice, it rescued the number of locomotor bouts and significantly improved bradykinesia. Additionally, we observed remodeling of the A13 connectivity post-nigrostriatal lesions. These findings highlight the A13 region as a key area involved in locomotor control and suggest a potential therapeutic target for PD-related motor deficits.

### The role of the A13 region in locomotion in sham mice

This study provides direct evidence that photoactivation of the A13 region can drive locomotion, suggesting that the pro-locomotor functions of the ZI extend further rostrally in the mouse. It adds to our work in rats where DBS of the A13 region evoked robust locomotor behavior (*Bisht et al., 2025*). Previous research has shown that photoactivation of caudal zona incerta (cZI) neurons increases animal movement speed in prey capture (*Zhao et al., 2019*) and active avoidance (*Hormigo et al., 2020*). Interestingly, early research on the cat suggested a role for the ZI in locomotion, particularly in the region adjacent to the STN, but the effectiveness of rostral ZI regions containing the A13 was not reported (*Grossman, 1958*). Recent work shows that the A13 is involved in forelimb grasping (*Garau*

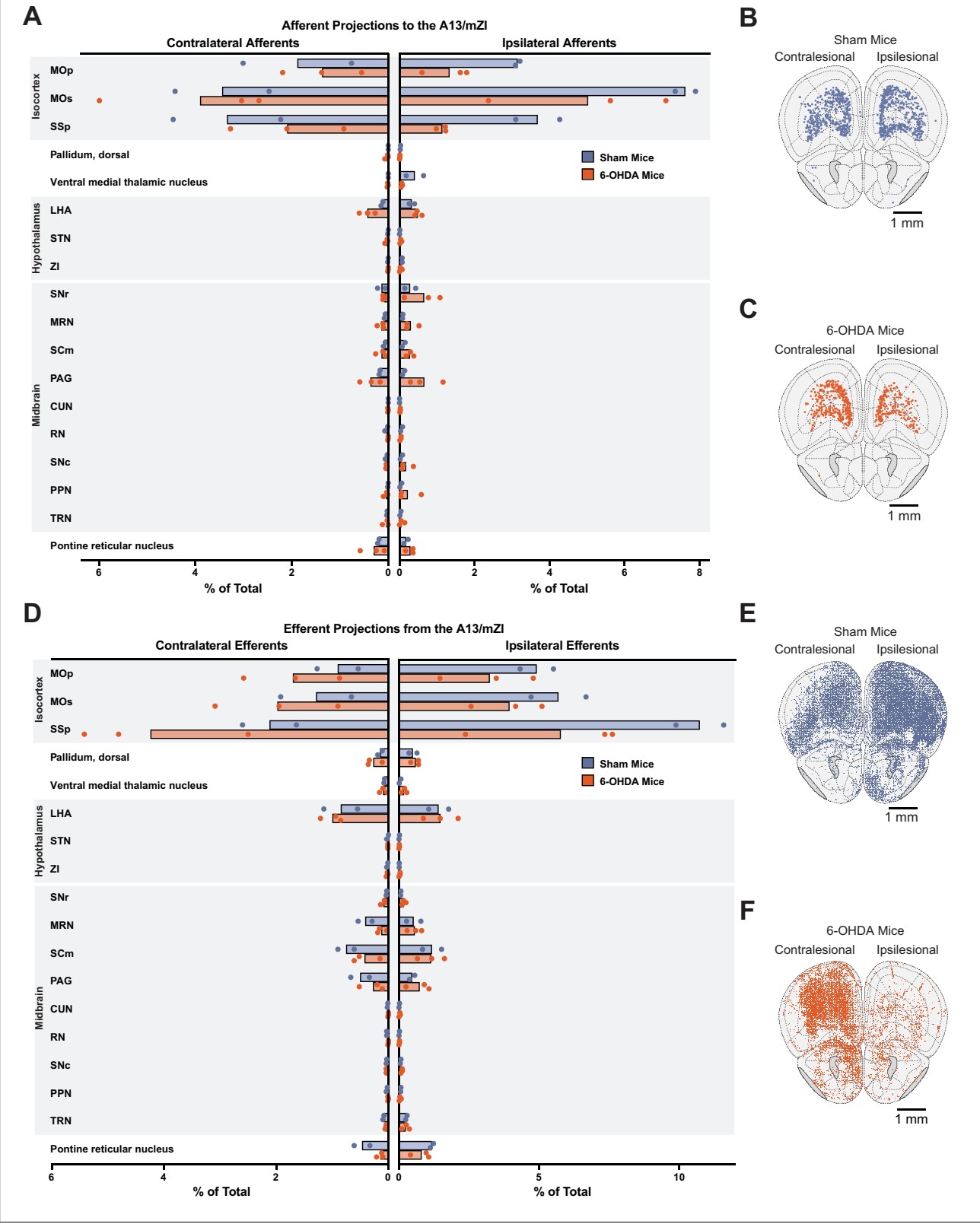

**Figure 6.** Unilateral nigrostriatal degeneration causes distinct changes in A13 connectivity. (**A**) Relative changes in A13 afferent (input) connections in 6-hydroxydopamine (6-OHDA)-lesioned mice compared to sham controls. Brain regions showing A13 input connections in sham (**B**) and 6-OHDA-lesioned (**C**) mice. (**D**) Relative changes in A13 efferent (output) connections in 6-OHDA-lesioned mice compared to sham controls. Brain regions showing A13 output connections in sham (**E**) and 6-OHDA-lesioned (**F**) mice. 6-OHDA: *n* = 3; sham: *n* = 2. Brain region abbreviations follow the

*Figure 6 continued on next page*

*Figure 6 continued*

Allen Brain Atlas: MOp (primary motor cortex), MOs (secondary motor cortex), SSp (primary somatosensory area), LHA (lateral hypothalamus), STN (subthalamic nucleus), ZI (zona incerta), SNr (substantia nigra pars reticulata), MRN (midbrain reticular nucleus), SCm (superior colliculus, motor), PAG (periaqueductal gray), CUN (cuneiform nucleus), RN (red nucleus), SNc (substantia nigra pars compacta), PPN (pedunculopontine nucleus), and TRN (tegmental reticular nucleus).

The online version of this article includes the following source data and figure supplement(s) for figure 6:

**Source data 1.** Normalized afferent and efferent cell counts in 6-hydroxydopamine (6-OHDA) and sham animals.

**Figure supplement 1.** Examples of retrogradely labeled GFP-positive fibers and cells from selected regions illustrating projections to the A13 region.

**Figure supplement 2.** Examples of anterogradely labeled mCherry-positive fibers from selected regions illustrating projections to the A13 region.

*et al., 2023*), suggesting other motor control functions, while a positive valence associated with motivated food seeking behavior has also been reported (*Ye et al., 2023*). Previous work targeting the mZI region, including somatostatin (SOM$^+$), calretinin (CR$^+$), and vGlut2$^+$ neurons, did not change locomotor distance traveled in the OFT (*Li et al., 2021*). However, multiple populations being photostimulated or targeting more medial populations in the ZI may contribute to the differences. Our findings align with mini-endoscope recordings from CaMKIIα$^+$ rostral ZI cells, which overlap the A13 showing subpopulations whose activity correlates with either movement speed or anxiety-related locations (*Li et al., 2021*). Other work has found that photoactivation of GABAergic mZI pathways, which project to the cuneiform, promotes exploratory activity by inhibiting cuneiform vGlut2 neurons (*Sharma et al., 2024*). There was a clear difference between the pro-locomotor patterns observed after general mZI activation in this study and those resulting from the activation of mZI GABA populations. The activity patterns with CamII kinase promotor transfection of the region produced a marked effect on locomotor speed, accompanied by thigmotactic behaviors not observed when mZI GABAergic populations are activated. The pro-locomotor effects observed from the mZI region, both in this study and others, differ from those seen when lateral GABAergic ZI populations (dorsal and ventral ZI) are stimulated. Microinjection of GABA$_A$ receptor agonists into the ZI significantly reduces locomotor distance and velocity and may induce cataplexy (*Chen et al., 2023*; *Wardas et al., 1988*). Alternatively, while suppressing GABAergic ZI activity with GABA$_A$ receptor antagonists can increase locomotion (*Périer et al., 2002*), chemogenetic or optogenetic inhibition in healthy naive mice can induce bradykinesia and akinesia (*Chen et al., 2023*). These contrasting outcomes likely stem from the differing projection patterns of GABAergic populations.

The increased locomotor speed and improved descent times on the pole test, resulting from A13 region photoactivation, highlight its role in movement control. Given that A13 stimulation did not alter coordination during the task, it suggests a complex behavioral role consistent with its upstream location from the brainstem and its extensive afferent and efferent projections. Notably, A13 photoactivation also increased animal speed, duration, and distance traveled. Collectively, these findings represent a rescue of function in the 6-OHDA model. Interestingly, both 6-OHDA and sham mice exhibited a latency of 10–15 s on average following photoactivation before locomotion was initiated. Such delays are typical when stimulating sites upstream of the cuneiform, such as the dlPAG, which shows delays of several seconds (*Tsang et al., 2021*).

## Photoactivation of the A13 reduces bradykinesia and akinesia in mouse PD models

While much work has targeted basal ganglia structures to address PD symptoms (*DeLong and Wichmann, 2015*), our research demonstrates that photoactivation of the A13 region can alleviate both bradykinesia and akinesia in 6-OHDA mice. Our work shows that A13 projections are affected at cortical and striatal levels following 6-OHDA, consistent with our observed changes in locomotor function. Over 28 days, there was a remarkable change in the afferent and efferent A13 connectome, despite the preservation of TH$^+$ ZI cells. This is consistent with previous reports of widespread connectivity of the ZI (*Mitrofanis, 2005*). The preservation of A13 is expected since A13 lacks DAT expression (*Bolton et al., 2015*; *Negishi et al., 2020*; *Sharma et al., 2018*) and is spared from DAT-mediated toxicity of 6-OHDA (*Dauer and Przedborski, 2003*; *Konnova, 2018*; *Simola et al., 2007*). While A13 cells were spared following nigrostriatal degeneration, our work demonstrates its connectome was rewired. The ipsilateral afferent projections were markedly downregulated, while contralesional

projecting afferents showed upregulation. In contrast, efferent projections showed less downregulation in the cortical subplate regions and bilateral upregulation of thalamic and hypothalamic efferents. Similar timeframes for anatomical and functional plasticity affecting neurons and astrocytes following an SNc or mFB 6-OHDA have been previously reported (*Bosson et al., 2015*; *Perović et al., 2005*; *Requejo et al., 2020*). Human PD brains that show degeneration of the SNc have a preserved A13 region, suggesting that our model, from this perspective, is externally valid (*Matzuk and Saper, 1985*).

Combined with photoactivation of the A13 region, we provide evidence for plasticity following damage to SNc. A previous brain-wide quantification of TH levels in the MPTP mouse model identified additional complexity in regulating central TH expression compared to conventional histological studies (*Roostalu et al., 2019*). These authors reported decreased SNc TH$^+$ cell numbers without a significant change in TH$^+$ intensity in SNc and increased TH$^+$ intensity in limbic regions such as the amygdala and hypothalamus (*Roostalu et al., 2019*). Still, there was a downstream shift in the distribution pattern of A13 efferents following nigrostriatal degeneration with a reduction in outputs to cortical and striatal subregions. This suggests A13 efferents are more distributed across the neuraxis than in sham mice. One hypothesis arising from our work is that the preserved A13 efferents could provide compensatory innervation with collateralization mediated contralesionally and, in some subregions, ipsilesionally to increase the availability of extracellular dopamine.

Several A13 efferent targets could be responsible for rotational asymmetry. In a unilateral 6-OHDA model, ipsiversive circling behavior is indicative of intact striatal function on the contralesional side (*Carey, 1991*; *Schwarting et al., 1991*; *Ungerstedt, 1971*; *Zetterström et al., 1986*). Instead, the predictive value of a treatment is determined by contraversive circling mediated by increased dopamine receptor sensitivity on the ipsilesional striatal terminals (*Costall et al., 1976*; *Lane et al., 2006*). Our findings show that A13 stimulation enhances ipsiversive circling and may represent a gain of function on the intact side, but this may be simply due to 6-OHDA mice having reduced locomotion overall. Given the preservation of A13 cells in PD, bilateral stimulation of A13 could potentially reduce motor asymmetry and alleviate bradykinesia and akinesia.

With the induction of a 6-OHDA lesion, there is a change in the A13 connectome, characterized by a reduction in bidirectional connectivity with ipsilesional cortical regions. In rodent models, the motor cortices, including the M1 and M2 regions, can shape rotational asymmetry (*Gradinaru et al., 2009*; *Magno et al., 2019*; *Sanders and Jaeger, 2016*; *Valverde et al., 2020*). Activation of M1 glutamatergic neurons increases the rotational bias (*Valverde et al., 2020*), while M2 neuronal stimulation promotes contraversive rotations (*Magno et al., 2019*). Our data suggest that A13 photoactivation may have resulted in the inhibition of glutamatergic neurons in the contralesional M1. An alternative possibility is the activation of the contralesional M2 glutamatergic neurons, which would be expected to induce increased ipsilesional rotations (*Magno et al., 2019*). The ZI could generate rotational bias by A13 modulation of cZI glutamatergic neurons via incerto-incertal fibers (*Ossowska, 2020*; *Power and Mitrofanis, 1999*), which promotes asymmetries by activating the SNr (*Li et al., 2022*). The incerto-incertal interconnectivity has not been well studied, but the ZI has a large degree of interconnectivity (*Sharma et al., 2018*; *Tsang et al., 2021*) along all axes and between hemispheres (*Power and Mitrofanis, 1999*). However, this may only contribute minimally given that unilateral photoactivation of the A13 cells in sham mice failed to produce ipsiversive turning behavior, while unilateral photoactivation of cZI glutamatergic neurons in sham animals was sufficient in generating ipsiversive turning behavior (*Li et al., 2022*). Another possibility involves the A13 region projections to the MLR. With the unknown downstream effects of A13 photoactivation, there may be modulation of the PPN neurons responsible for this turning behavior (*Masini and Kiehn, 2022*). The thigmotactic behaviors suggest some effects may be mediated through dlPAG and CUN (*Tsang et al., 2021*), and recent work suggests the CUN as a possible therapeutic target (*Fougère et al., 2021*; *Noga and Whelan, 2022*). Since PD is a heterogeneous disease, our data provide another therapeutic target providing context-dependent relief from symptoms. This is important since PD severity, symptoms, and progression are patient specific.

## Toward a preclinical model

To facilitate future translational work applying DBS to this region, we targeted the A13 region using AAV8-CamKII-mCherry viruses. The CaMKIIα promoter virus is beneficial because it is biased toward excitatory cells (*Haery et al., 2019*), narrowing the diversity of the transfected A13 region, and when

combined with traditional therapies, such as L-DOPA, it could be a translatable strategy (*Watakabe et al., 2015*; *Watanabe et al., 2020*). Optogenetic strategies have been used to activate retinal cells in humans, partially restoring visual function and providing optimism that AAV-based viral strategies can be adapted in other human brain regions (*Sahel et al., 2021*). A more likely possibility for stimulation of deep nuclei is that DREADD technology could be adapted, which would not require any implants; however, this remains speculative. Our recent work demonstrates that the A13 is a target for DBS, where stimulation in rats, as predicted, produced robust increases in locomotor activity (*Bisht et al., 2025*). Gait dysfunction in PD is particularly difficult to treat, and indeed when DBS of the STN is deployed, a mixture of unilateral and bilateral approaches has been used (*Lizarraga et al., 2016*), along with stimulation of multiple targets (*Stefani et al., 2007*). This represents the heterogeneity of PD and underlines the need for considering multiple targets. DBS does not always have the same outcomes as optogenetic stimulation (*Neumann et al., 2023*), and our DBS shows a blend of anxiolytic and pro-locomotory effects, as predicted by this work and our work activating GABAergic mZI populations (*Bisht et al., 2025*; *Sharma et al., 2024*). Future work may want to consider a multipronged strategy to hone burst stimulation parameters with identification of cell populations to deploy DBS in a more targeted manner (*Spix et al., 2021*).

## Limitations

Currently, few PD animal models are available that adequately model the progression and the extent of SNc cellular degeneration while meeting the face validity of motor deficits (*Dauer and Przedborski, 2003*; *Konnova, 2018*). While the 6-OHDA models fail to capture the age-dependent chronic degeneration observed in PD, they lead to robust motor deficits with acute degeneration and allow for compensatory changes in connectivity to be examined. Moreover, the 6-OHDA lesions resemble the unilateral onset (*Hughes et al., 1992*) and persistent asymmetry (*Lee et al., 1995*) of motor dysfunction in PD. Another option could be the MPTP mouse model, which offers the ease of systemic administration and translational value to primate models; however, the motor deficits are variable and lack the asymmetry observed in human patients (*Hughes et al., 1992*; *Jagmag et al., 2015*; *Lee et al., 1995*; *Meredith and Rademacher, 2011*). Despite these limitations, the neurotoxin-based mouse models, such as MPTP and 6-OHDA, offer greater SNc cell loss than genetic-based models; in the case of the 6-OHDA model, it captures many aspects of motor dysfunctions in PD (*Dauer and Przedborski, 2003*; *Jagmag et al., 2015*; *Konnova, 2018*; *Simola et al., 2007*). As in human PD, we found no significant change in A13 TH$^+$ cell counts (*Matzuk and Saper, 1985*). Another limitation is that since A13 neurons remained intact following a lesion, it is possible that changes in the connectome reflected secondary effects from other regions impacted by the 6-OHDA lesion. However, the fact that there was a significant change in the connectome post-6-OHDA injection and striatonigral degeneration is in and of itself important to document. Finally, it is important to note that our whole-brain anatomical data offer a correlative framework for understanding the neural circuits involved in A13-mediated locomotor control and its modulation in the 6-OHDA model. However, these data do not establish direct causal relationships. Future studies employing techniques such as targeted pathway manipulations (e.g., optogenetics and chemogenetics) or lesioning will be essential to definitively prove the functional necessity of specific connections in mediating the observed behavioral effects. A primary limitation of our whole-brain connectomic screen is the small sample size. This restricts the statistical power of our comparisons, and the imaging should be viewed as a preliminary, exploratory screen that provides valuable initial insights into the potential reorganization of the A13 connectome in the 6-OHDA model. Future studies with larger cohorts will be essential to confirm these findings. That said, the global approach allows us to identify widespread changes in connectivity that might be overlooked by more targeted analyses, offering insights into the complex neural adaptations that occur following nigrostriatal degeneration.

## Conclusions

Our research underscores the role of the A13 region beyond the classic nigrostriatal axis in PD, driving locomotor activity and mitigating bradykinetic and akinetic deficits linked to impaired DAergic transmission. This observation indicates a rescue of locomotion loss in 6-OHDA-lesioned mice, as well as bradykinesia. Additionally, it produced possible gain-of-function effects, such as circling behavior, which may be attributed to plasticity changes induced by the 6-OHDA lesions. Widespread

remodeling of the A13 region connectome is critical to our understanding of the effects of dopamine loss in PD models. In summary, our findings support an exciting role for the A13 region in locomotion with demonstrated benefits in a mouse PD model and contribute to our understanding of heterogeneity in PD.

# Materials and methods

## Key resources table

| Reagent type (species) or resource | Designation | Source or reference | Identifiers | Additional information |
|---|---|---|---|---|
| strain, strain background (C57Bl/6 mice) | C57 | Charles River | C57BL/6NCrl, RRID:IMSR_CRL:027 | |
| antibody | anti-cFos (Rabbit polyclonal) | Synaptic Systems | Cat# 226 003, RRID:AB_2231974 | IF(1:1000), A13 region |
| antibody | anti-GFP (Chicken polyclonal) | Aves Lab | Cat# GFP-1010, RRID:AB_2307313 | IF(1:1000), whole brain; IF(1:5000), A13 region |
| antibody | anti-mCherry (Rat monoclonal) | Invitrogen, Thermo Fisher Scientific | Cat# M11217; RRID:AB_2536611 | IF(1:500), whole brain |
| antibody | anti-Tyrosine Hydroxylase (Rabbit polyclonal) | Abcam | Cat# AB-112, RRID:AB_2307313 | IF(1:500), whole brain; IF(1:1000), SNc region |
| antibody | anti-Tyrosine Hydroxylase (Sheep polyclonal) | Millipore Sigma | Cat# AB1542; RRID:AB_90755 | IF(1:500), A13 region |
| antibody | Alexa Fluor 488 Donkey Anti-Chicken | JacksonImmuno | 703–545- 155, RRID:AB_2340375 | IF(1:200), whole brain; IF(1:1000) A13 region |
| antibody | Alexa Fluor 594 Donkey Anti-Rabbit | Invitrogen, Thermo Fisher Scientific | A-21207, RRID:AB_3695597 | IF(1:500), A13 region |
| antibody | Alexa Fluor 647 Donkey Anti-Rabbit | Invitrogen, Thermo Fisher Scientific | A-31573, RRID:AB_2536183 | IF(1:1000), SNc region |
| antibody | Alexa Fluor 647 Donkey Anti-Sheep | Invitrogen, Thermo Fisher Scientific | A-21448, RRID:AB_2535865 | IF(1:1000), A13 region |
| antibody | Alexa Fluor 790 Donkey Anti-Rabbit | Invitrogen, Thermo Fisher Scientific | A-11374, RRID:AB_2534145 | IF(1:200), whole brain |
| antibody | Cy3 AffiniPure Donkey Anti-Rat | JacksonImmuno | 712-165-153, RRID:AB_2340667 | IF(1:200), whole brain |
| recombinant DNA reagent | AAV8-CamKII-mCherry | Neurophotonics | Cat# KD8-aav1 | Lot #820, titre 2×1,013 GC/ml |
| recombinant DNA reagent | AAVrg-CAG- GFP | Addgene | Cat# 37825, RRID:Addgene_37825 | Lot #V9234, titre ≥7 × 10¹² vg/mL |
| recombinant DNA reagent | AAVDJ-CaMKIIα-hChR2(H134R)-eYFP | UNC Stanford Viral Vector Core | Cat# AAV36 | Lots #3081 and #6878, titres 1.9×1,013 and 1.7×1,013 GC/mL |
| recombinant DNA reagent | AAVDJ-CaMKIIα-eYFP | UNC Stanford Viral Vector Core | Cat# AAV08 | Lots #2958 and #5510, titres 7.64×1,013 and 2.88×1,013 GC/mL |
| chemical compound, drug | Desipramine hydrochloride | Sigma Aldrich | D3900 | 2.5 mg/mL |
| chemical compound, drug | Pargyline hydrochloride | Sigma Aldrich | P8013 | 0.5 mg/mL |
| chemical compound, drug | 6-OHDA | Tocris | 2547/50 | 15.0 mg/mL |
| software, algorithm | Adobe Illustrator | Adobe | RRID:SCR_010279 | |
| software, algorithm | ImageJ | ImageJ | RRID:SCR_003070 | |
| software, algorithm | WholeBrain | WholeBrain | RRID:SCR_015245 | |
| software, algorithm | Prism | GraphPad | RRID:SCR_002798 | |
| software, algorithm | SPSS | SPSS | RRID:SCR_002865 | |
| other | DAPI stain | Invitrogen, Thermo Fisher Scientific | D1306, RRID:AB_2629482 | IF(1:1000), A13/SNc regions |
| other | TO-PRO–3 Iodide (642/661) stain | Invitrogen, Thermo Fisher Scientific | T3605 | IF(1:5000), whole brain |

## Animals

All care and experimental procedures were approved by the University of Calgary Health Sciences Animal Care Committee (Protocol #AC19-0035). C57BL/6 male mice 49–56 days old (weight: $M$ = 31.7 g, SEM = 2.0 g) were group-housed (<five per cage) on a 12-hr light/dark cycle (07:00 lights on – 19:00 lights off) with ad libitum access to food and water, as well as cat's milk (Whiskas, Mars Canada Inc, Bolton, ON, Canada). Mice were randomly assigned to the groups described.

## Surgical procedures

We established a well-validated unilateral 6-OHDA-mediated Parkinsonian mouse model (*Thiele et al., 2012*; *Figure 1*). Thirty minutes before stereotaxic microinjections, mice were intraperitoneally injected with desipramine hydrochloride (2.5 mg/ml, Sigma-Aldrich) and pargyline hydrochloride (0.5 mg/ml, Sigma-Aldrich) at 10 ml/kg (0.9% sterile saline, pH 7.4) to enhance selectivity and efficacy of 6-OHDA induced lesions (*Thiele et al., 2012*). All surgical procedures were performed using aseptic techniques, and mice were anesthetized using isoflurane (1–2%) delivered by 0.4 l/min of medical-grade oxygen (Vitalair 1072, 100% oxygen).

Mice were stabilized on a stereotaxic apparatus. Small craniotomies were made above the mFB and the A13 nucleus within one randomly assigned hemisphere. Stereotaxic microinjections were performed using a glass capillary (Drummond Scientific, PA, USA; Puller Narishige, diameter 15–20 mm) and a Nanoject II apparatus (Drummond Scientific, PA, USA). 240 nl of 6-OHDA (3.6 µg, 15.0 mg/ml; Tocris, USA) was microinjected into the MFB (AP –1.2 mm from bregma; ML ±1.1 mm; DV –5.0 mm from the dura). Sham mice received a vehicle solution (240 nl of 0.2% ascorbic acid in 0.9% saline; Tocris, USA).

### Whole-brain experiments

For tracing purposes, a 50:50 mix of AAV8-CamKIIα-mCherry (Neurophotonics, Laval University, Quebec City, Canada, Lot #820, titer $2 \times 10^{13}$ GC/ml) and AAVrg-CAG-GFP (Addgene, Watertown, MA, Catalogue #37825, Lot #V9234, titer $\geq 7 \times 10^{12}$ vg/ml) was injected ipsilateral to 6-OHDA injections at the A13 nucleus in all mice (AP –1.22 mm from bregma; ML ±0.4 mm; DV –4.5 mm from the dura, the total volume of 110 nl at a rate of 23 nl/s). Post-surgery care was the same for both sham and 6-OHDA-injected mice. The animals were sacrificed 29 days after surgery.

### Photoactivation experiments

36.8 nl of AAVDJ-CaMKIIα-hChR2(H134R)-eYFP (UNC Stanford Viral Gene Core; Stanford, CA, US, Catalogue #AAV36; Lots #3081 and #6878, titers $1.9 \times 10^{13}$ and $1.7 \times 10^{13}$ GC/ml, respectively) or eYFP control virus (AAVDJ-CaMKIIα-eYFP; Catalogue #AAV08; Lots #2958 and #5510, titers $7.64 \times 10^{13}$ and $2.88 \times 10^{13}$ GC/ml, respectively) were injected into the A13 (AP: –1.22 mm; ML –0.5 mm from the Bregma; DV –4.5 mm from the dura). A mono-fiber cannula (Doric Lenses, Quebec, Canada, Catalogue #B280-2401-5, MFC_200/230–0.48_5mm_MF2.5_FLT) was implanted slowly 300 µm above the viral injection site. Metabond Quick Adhesive Cement System (C&B, Parkell, Brentwood, NY, US) and Dentsply Repair Material (Dentsply International Inc, York, PA, USA) were used to fix the optical fiber in place. Animals recovered from the viral surgery for 19 days before follow-up behavioral testing. *Figure 1* shows a timeline of behavioral tests.

## ChR2 photoactivation

Photoactivation was achieved using a 473-nm laser and driver (LRS-0473-GFM-00100-05, Laserglow Technologies, North York, ON, Canada). The laser was triggered by TTL pulses from an A.M.P.I. Master-8 stimulator (Jerusalem, Israel) or an Open Ephys PulsePal (Sanworks, Rochester, NY, US) set to 20 Hz, 10 ms pulse width, and 5 mW power. All fiber optic implants were tested for laser power before implantation (Thorlabs, Saint-Laurent, QC, Canada; optical power sensor (S130C) and meter (PM100D)). The Stanford Optogenetics irradiance calculator was used to estimate the laser power for stimulation (*Stanford Optogenetics Resource Center, 2020*). A 1 × 2 fiber-optic rotary joint (Doric Lenses, Quebec, Canada; FRJ_1x2i_FC-2FC_0.22) was used. The animals' behaviors were recorded with an overhead camera (SuperCircuits, Austin, TX, US; FRJ_1x2i_FC-2FC_0.22; 720x480 resolution; 30 fps). The video was processed online (Cleversys, Reston, VA; TopScan V3.0) with a TTL signal

output from a National Instruments 24-line digital I/O box (NI, Austin, TX, US; USB-6501) to the Master-8 stimulator.

## Behavioral testing

### Open-field test

Each mouse was placed in a square arena measuring 70 (W) × 70 (L) × 50 (H) cm with opaque walls and recorded for 30 min using a vertically mounted video camera (Model PC165DNR, Supercircuits, Austin, TX, USA; 30 fps). 19 days following surgery, mice were habituated to the OFT arena with a patch cable attached for 3 days in 30-min sessions to bring animals to a common baseline of activity. On experimental days, after animals were placed in the OFT, a one-minute-on-three-minutes-off paradigm was repeated five times following an initial 10 min baseline activity. Locomotion was registered when mice traveled a minimum distance of 10 cm at 6 cm/s for 20 frames over a 30-frame segment. When the mouse velocity dropped below 6 cm/s for 20 frames, locomotion was recorded as ending. Bouts of locomotion relate to the number of episodes where the animal met these criteria. Velocity data were obtained from the frame-by-frame results and further processed in a custom Python script to detect instantaneous speeds greater than 2 cm/s (*Masini and Kiehn, 2022*). All animals that had validated targeting of the A13 region were included in the OFT data presented in the results section, except for one sham ChR2 animal, which showed grooming rather than the typical locomotor phenotypes.

### Pole test

Mice were placed on a vertical wooden pole (50 cm tall and 1 cm diameter) facing upwards and then allowed to descend the pole into their home cage (*Glajch et al., 2012*). Animals were trained for 3 days and tested 2–5 days pre-surgery. Animals were acclimatized 21–22 days post-surgery under two conditions: without a patch cable and with the patch cable attached without photoactivation. On days 24–27, experimental trials were recorded with photoactivation. Video data were recorded for a minimum of three trials (Canon, Brampton, ON, Canada; Vixia HF R52; 1920 × 1080 resolution; 60 fps). A blinded scorer recorded the times for the following events: the hand release of the animal's tail, the animal fully turning to descend the pole, and the animal reaching the base of the apparatus. Additionally, partial falls, where the animal slipped down the pole but did not reach the base, and full falls, where the animal fell to the base, were recorded separately. All validated animals were included in the quantified data, including the sham ChR2 animal that began grooming in the OFT upon photoactivation. This animal displayed proficiency in performing the pole test during photoactivation. It started grooming upon completion of the task when photoactivation was on. One sham ChR2 animal was photostimulated at 1 mW since it would jump off the apparatus at higher stimulation intensities.

## Immunohistochemistry

### A13 and SNc region

Post hoc analysis of the tissue was performed to confirm the 6-OHDA lesion and validate the targeting of the A13 region. Following behavioral testing, animals underwent a photoactivation protocol to activate neurons below the fiber optic tip (*Koblinger et al., 2018*). Animals were placed in an OFT for 10 min before receiving 3 min of photoactivation. Ten minutes later, the animals were returned to their home cage. Ninety minutes post photoactivation, animals were deeply anesthetized with isoflurane and then transcardially perfused with room temperature PBS followed by cold 4% paraformaldehyde (PFA) (Sigma-Aldrich, Catalogue #441244-1KG). The animals were decapitated, and the whole heads were incubated overnight in 4% PFA at 4°C before the fiber optic was removed and the brain was removed from the skull. The brain tissue was post-fixed for another 6–12 hr in 4% PFA at 4°C then transferred to 30% sucrose solution for 48–72 hr. The tissue was embedded in VWR Clear Frozen Section Compound (VWR International LLC, Radnor, PA, US) and sectioned coronally at 40 or 50 µm using a Leica cryostat set to –21°C (CM 1850 UV, Concord, ON, Canada). Sections from the A13 region (−0.2 to –2.0 mm past bregma) and the SNc (−2.2 to –4.0 mm past bregma) were collected and stored in PBS containing 0.02% (wt/vol) sodium azide (EM Science, Catalogue #SX0299-1, Cherry Hill, NJ, US) (*Paxinos and Franklin, 2008*).

Immunohistochemistry staining was done on free-floating sections. The A13 sections were labeled for c-Fos, TH, and GFP (to enhance eYFP viral signal), and received a DAPI stain to identify nuclei. The SNc sections were stained with TH and DAPI. Sections were washed in PBS (3 × 10 min) then

**Table 1.** Whole-brain clearing protocol.

| Day # | Instructions |
|---|---|
| 1 | Washed 2 × 30 min in PBS on a shaker at room temperature. |
| 2 | Dehydrated tissue in methanol/H$_2$O series of 20%, 40%, 60%, 80%, 100%, 100% (1 hr each at room temperature) then left overnight in a 66% dichloromethane/33% methanol solution. |
| 3 | Washed twice in 100% methanol at room temperature and then chilled the samples at 4°C. Bleached the samples in chilled fresh 5% H$_2$O$_2$ in methanol (1 volume 30% H$_2$O$_2$ to 5 volumes methanol), overnight at 4°C. |
| 4 | Rehydrated in methanol/water series of 80%, 60%, 40%, 20%, PBS (1 hr each at room temperature). Washed twice in PTx.2 (0.2% Triton X-100 in PBS). Incubated the samples in Permeabilization Solution at 37°C for 2 days. |
| 6 | Washed 3 × 1–2 hr in 0.5 mM SDS/PBS at 37°C then incubated for 3 days. |
| 9 | Incubated with the primary antibody in 0.5 mM SDS/PBS at 37°C for 3 days. |
| 12 | Refreshed with the primary antibody in PTx.2 at 37°C then incubated for 4 days. |
| 17 | Washed 5 × 2 hr in PTwH (0.2% Tween 20 and 10 mg/ml Heparin stock solution in PBS) and incubated at 37°C overnight. |
| 18 | Incubated with secondary antibody in PTwH/3% Donkey Serum at 37°C for 3 days. |
| 21 | Refreshed with secondary antibody in PTwH/3% Donkey Serum at 37°C for 4 days. |
| 25 | Washed 5 × 2 hr in PTwH then incubated at 37°C overnight. |
| 26 | Dehydrated in methanol/water series of 20%, 40%, 60%, 80%, 100% (1 hr each at room temperature). Refreshed the samples with 100% methanol and left overnight at room temperature. |
| 27 | Incubated in 66% DCM/33% methanol for 3 hr on a shaker at room temperature. Then incubated in 100% DCM (Sigma 270997-12X100 ML) for 2 × 15 min (with shaking) to wash away methanol. Incubated samples with ethyl cinnamate for 3 hr at room temperature with shaking. Refreshed ethyl cinnamate, then left at room temperature for imaging. |

incubated in a blocking solution comprised of PBS containing 0.5% Triton X-100 (Sigma-Aldrich, Catalogue #X100-500ML, St. Louis, MO, US) and 5% donkey serum (EMD Millipore, Catalogue #S30-100ML, Billerica, MA, USA) for 1 hr. This was followed by overnight (for SNc sections) or 24 hr (for A13 sections) incubation in a 5% donkey serum PBS primary solution at room temperature. On day 2, the tissue was washed in PBS (3 × 10 min) before being incubated in a PBS secondary solution containing 5% donkey serum for 2 hr (for SNc tissue) or 4 hr (for A13 tissue). The secondary was washed with a PBS solution containing 1:1000 DAPI for 10 min, followed by a final set of PBS washes (3 × 10 min). Tissue was mounted on Superfrost micro slides (VWR, slides, Radnor, PA, US) with mounting media (Vectashield, Vector Laboratories Inc, Burlingame, CA, US), covered with #1 coverslips (VWR, Radnor, PA, US), then sealed.

### Whole brain

Mice were deeply anesthetized with isoflurane and transcardially perfused with PBS, followed by 4% PFA. To prepare for whole-brain imaging, brains were first extracted and postfixed overnight in 4% PFA (*Table 1*) at 4°C. The next day, a modified iDISCO method (*Renier et al., 2014*) was used to clear the samples and perform quadruple immunohistochemistry in whole brains. The modifications include prolonged incubation and the addition of SDS for optimal labeling. Antibodies used are listed in the Key Resources Table and the protocol is provided in the Whole Brain Clearing Protocol.

## Image acquisition and analysis

### Photoactivation experiments

All tissue was initially scanned with an Olympus VS120-L100 Virtual Slide Microscope (UPlanSApo, 10x and 20x, NA = 0.4 and 0.75). Standard excitation and emission filter cube sets were used (DAPI, FITC, TRITC, and Cy5), and images were acquired using an Orca Flash 4.0 sCMOS monochrome camera (Hamamatsu, Bridgewater Township, NJ, US). For c-Fos immunofluorescence, A13 sections of the tissue were imaged with a Leica SP8 FALCON (FAst Lifetime CONtrast) scanning confocal microscope equipped with a tunable laser and using a 63x objective (HC PlanApo, NA = 1.40).

SNc images were imported into Adobe Illustrator, where the SNc (*Fougère et al., 2021*), including the pars lateralis (SNl), was delineated using the TH immunostaining together with the medial lemniscus

and cerebral peduncle as landmarks (bregma –3.09 and –3.68 mm) (*Iancu et al., 2005*; *Paxinos and Franklin, 2008*; *Stott and Barker, 2014*). Cell counts were obtained using a semi-automated approach using an Ilastik (v1.4.0b15) (*Berg et al., 2019*) trained model followed by corrections by a blinded counter (*Fougère et al., 2021*; *Iancu et al., 2005*). Targeting was confirmed on the 10x overview scans of the A13 region tissue by the presence of eYFP localized in the mZI around the A13 TH$^+$ nucleus, the fiber optic tip being visible near the mZI and A13 nucleus, and the presence of c-Fos positive cells in ChR2$^+$ tissue. C-Fos expression colocalization within the A13 region was performed using confocal images. The mZI and A13 region was identified with the 3rd ventricle and TH expression as markers (*Paxinos and Franklin, 2008*).

## Whole-brain experiments

Cleared whole-brain samples were imaged using a light-sheet microscope (LaVision Biotech UltraMicroscope, LaVision, Bielefeld, Germany) with an Olympus MVPLAPO 2x objective with 4x optical zoom (NA = 0.475) and a 5.7 mm dipping cap that is adjusted for the high refractive index of 1.56. The brain samples were imaged in an ethyl cinnamate medium to match the refractive indices and illuminated by three sheets of light bilaterally. Each light sheet was 5 µm thick, and the width was set at 30% to ensure sufficient illumination at the centroid of the sample. Laser power intensities and chromatic aberration corrections used for each laser were as follows: 10% power for 488 nm laser, 5% power for 561 nm laser with 780 nm correction, 40% power for 640 nm laser with 960 nm correction, and 100% power for 785 nm laser with 1620 nm correction. Each sample was imaged coronally in 8 by 6 squares with 20% overlap (10,202 µm by 5492 µm in total) and a z-step size of 15 µm (*xyz* resolution = 0.813 µm × 0.813 µm × 15 µm). While an excellent choice for our work, confocal microscopy offers better resolution at the expense of time. To gain a better resolution using a light-sheet microscope in select regions (eg. SNc and A13 cells), we increased the optical zoom to 6.3x.

## A13 connectome analysis

Images were processed using ImageJ software (*Schneider et al., 2012*). Raw images were stitched, and a z-encoded maximum intensity projection across a 90-µm-thick optical section was obtained across each brain. 90 µm sections were chosen because the 2008 Allen reference atlas images are spaced out at around 100 µm. Brains with insufficient quality in labeling were excluded from analysis (*n* = 1 of three sham and *n* = 3 of six 6-OHDA mice). Instructions for identifying YFP$^+$ or TH$^+$ cells to annotate were provided to the manual counters. YFP$^+$ and TH$^+$ cells were manually counted using the Cell Counter Plug-In (ImageJ). mCherry$^+$ fibers were segmented semi-automatically using Ilastik software (*Berg et al., 2019*) and quantified using particle analysis in ImageJ. Images and segmentations were imported into WholeBrain software to be registered with the 2008 Allen reference atlas (*Fürth et al., 2018*). The TO-PRO-3 and TH channels were used as reference channels to register each section to a corresponding atlas image. ImageJ quantifications of cell and fiber segmentations were exported in XML formats and registered using WholeBrain software. To minimize the influence of experimental variation on the total labeling of neurons and fibers, the afferent cell counts or efferent fiber areas in each brain region were column divided by the total number found in a brain to obtain the proportion of total inputs and outputs. Connectome analyses were performed using custom R scripts. For interregional correlation analyses, the data were normalized to a log$_{10}$ value to reduce variability and bring brain regions with high and low proportions of cells and fibers to a similar scale. The consistency of afferent and efferent proportions between mice was compared in a pairwise manner using Spearman's correlation (*Figure 6—figure supplement 1*).

## Quantification of 6-OHDA-mediated TH$^+$ cell loss

The percentage of TH$^+$ cell loss was quantified to confirm 6-OHDA-mediated SNc lesions. TH$^+$ cells within ZI, VTA, and SNc areas from 90 µm thick optical brain slice images (AP: –0.655 to –3.88 mm from bregma) were manually counted by two blinded counters (*n* = 3 sham and *n* = 6 6-OHDA mice; ZI region in 2 of 6 6-OHDA mice was excluded due to presence of abnormal scarring/healing at the injection site of viruses). Subsequently, WholeBrain software (*Fürth et al., 2018*) was used to register and tabulate TH$^+$ cells in the contralesional and ipsilesional brain regions of interest. Counts obtained from the two counters were averaged per region. The percentage of TH$^+$ cell loss was calculated by

dividing the difference in counts between contralesional and ipsilesional sides by the contralesional side count and multiplying by 100%.

## Statistical analyses

All data were tested for normality using a Shapiro–Wilk test to determine the most appropriate statistical tests. The percent ipsilesional TH+ neuron loss within the SNc, as defined above using a Pearson correlation (*Fougère et al., 2021*) was used to ascertain the effect of the 6-OHDA lesion on behavior. A Wilcoxon rank-sum test was performed for comparisons within subjects at two time points where normality failed, and the central limit theorem could not be applied. The two groups were compared using an unpaired *t*-test with Welch's correction. A mixed model ANOVA (MM ANOVA) was used to compare the effects of group type, injection type, and time. Additionally, Mauchly's test of sphericity was performed to account for differences in variability within the repeated measures design. A Greenhouse–Geisser correction was applied to all ANOVAs where Mauchly's test was significant for RM and MM ANOVAs. The post hoc multiple comparisons were run when the respective ANOVAs reached significance using Dunnett's or Dunn's tests for repeated measures of parametric and non-parametric tests, respectively. The pre-stimulation time points were used as the control time point to determine if stimulation altered behavior. A Bonferroni correction was added for post hoc comparisons following an MM ANOVA between groups at given time points to control for alpha value inflation. All correlations, *t*-tests, and ANOVAs were performed, and graphs were created using Prism version 9.3.1 (GraphPad) or SPSS (IBM, 28.0.1.0). Full statistical reporting is in Supplemental Statistics.xls.

## Acknowledgements

We would like to acknowledge support from Whelan and Kiss Labs and technical support from Hotchkiss Brain Institute Advanced Microscopy Platform Core Facility, Cumming School of Medicine Optogenetics Platform Core Facility and Drs. David Elliot, Jonathan Epp, Young Ou, and Lothar Resch. We acknowledge studentships from Parkinson Alberta (LHK), Parkinson Canada (LHK), Canadian Open Neuroscience Platform (AL), Cumming School of Medicine (AL, LHK), Faculty of Graduate Studies (AL, LHK), and the Faculty of Veterinary Medicine (CM, ST). This research was supported by grants to PJW provided by a Canadian Institutes of Health Research Project Grant (PJT-173511), Wings for Life, NSERC (RGPIN/04394-2019) as well as ZHTK from NSERC (RPGIN/04126-2017).

## Additional information

### Funding

| Funder | Grant reference number | Author |
| --- | --- | --- |
| Hotchkiss Brain Institute, University of Calgary | PFUN | Patrick J Whelan |
| Canadian Institutes of Health Research | PJT-173511 | Patrick J Whelan |
| Wings for Life | Project Research Grant | Patrick J Whelan |
| Natural Sciences and Engineering Research Council of Canada | RGPIN/04394-2019 | Patrick J Whelan |
| Natural Sciences and Engineering Research Council of Canada | RPGIN/04126-2017 | Zelma HT Kiss |
| Parkinson Canada | Graduate Studentship | Linda H Kim |
| Parkinson Alberta | Graduate Studentship | Linda H Kim |
| Canadian Open Neuroscience Platform | Graduate Studentship | Adam Lognon |

| Funder | Grant reference number | Author |
|---|---|---|
| Faculty of Veterinary Medicine, University of Calgary | Summer Studentship | Claire McPherson Stephanie Tam |

The funders had no role in study design, data collection, and interpretation, or the decision to submit the work for publication.

## Author contributions

Linda H Kim, Conceptualization, Data curation, Software, Formal analysis, Supervision, Investigation, Methodology, Writing – original draft, Project administration, Writing – review and editing; Adam Lognon, Conceptualization, Data curation, Formal analysis, Funding acquisition, Investigation, Methodology, Writing – original draft, Writing – review and editing; Sandeep Sharma, Stephanie Tam, Claire McPherson, Todd E Stang, Data curation, Investigation, Writing – review and editing; Michelle A Tran, Data curation, Formal analysis, Validation, Writing – review and editing; Cecilia Badenhorst, Data curation, Software, Formal analysis, Investigation, Writing – review and editing; Taylor Chomiak, Formal analysis, Validation, Visualization, Writing – review and editing; Shane EA Eaton, Data curation, Formal analysis, Investigation, Writing – review and editing; Zelma HT Kiss, Conceptualization, Supervision, Writing – review and editing; Patrick J Whelan, Conceptualization, Supervision, Funding acquisition, Writing – original draft, Project administration, Writing – review and editing

## Author ORCIDs

Linda H Kim ⓘ http://orcid.org/0000-0003-0628-3104
Sandeep Sharma ⓘ https://orcid.org/0000-0002-9680-7460
Michelle A Tran ⓘ http://orcid.org/0000-0002-2856-919X
Cecilia Badenhorst ⓘ http://orcid.org/0000-0003-2426-4783
Taylor Chomiak ⓘ https://orcid.org/0000-0001-6118-813X
Todd E Stang ⓘ https://orcid.org/0000-0003-3780-139X
Patrick J Whelan ⓘ https://orcid.org/0000-0002-1234-5415

## Ethics

All animal care and experimental procedures were approved by the University of Calgary Health Sciences Animal Care Committee (Protocol #AC19-0035).

Reviewer #1 (Public review): https://doi.org/10.7554/eLife.90832.4.sa1
Reviewer #2 (Public review): https://doi.org/10.7554/eLife.90832.4.sa2
Reviewer #3 (Public review): https://doi.org/10.7554/eLife.90832.4.sa3
Author response https://doi.org/10.7554/eLife.90832.4.sa4

# Additional files

## Supplementary files
MDAR checklist

## Data availability

All datasets and code have been deposited at Open Science Framework (OSF https://doi.org/10.17605/OSF.IO/J4AXW).

The following dataset was generated:

| Author(s) | Year | Dataset title | Dataset URL | Database and Identifier |
|---|---|---|---|---|
| Kim LH, Lognon A, Sharma S, Tran MA, Badenhorst C, Chomiak T, Tam S, McPherson C, Stang T, Eaton SEA, Kiss ZHT, Whelan PJ | 2021 | Restoration of locomotor function following stimulation of the A13 region in Parkinson's mouse models. | https://doi.org/10.17605/OSF.IO/J4AXW | Open Science Framework, 10.17605/OSF.IO/J4AXW |

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
