## [Editor Report · eLife Assessment]

This **valuable** study reveals the pro-locomotor effects of activating a deep brain region containing diverse range of neurons in both healthy and Parkinson's disease mouse models. While the findings are **solid**, mechanistic insights remain limited due to the small sample size. This research is relevant to motor control researchers and offers clinical perspectives.

---

## [Referee Report · Reviewer #1 (Public review)]

Summary:

This study aimed to investigate the effects of optically stimulating the A13 region in healthy mice and a unilateral 6-OHDA mouse model of Parkinson's disease (PD). The primary objectives were to assess changes in locomotion, motor behaviors, and the neural connectome. For this, the authors examined the dopaminergic loss induced by 6-OHDA lesioning. They found a significant loss of tyrosine hydroxylase (TH+) neurons in the substantia nigra pars compacta (SNc) while the dopaminergic cells in the A13 region were largely preserved. Then, they optically stimulated the A13 region using a viral vector to deliver the channelrhodopsine (CamKII promoter). In both sham and PD model mice, optogenetic stimulation of the A13 region induced pro-locomotor effects, including increased locomotion, more locomotion bouts, longer durations of locomotion, and higher movement speeds. Additionally, PD model mice exhibited increased ipsilesional turning during A13 region photoactivation. Lastly, the authors used whole-brain imaging to explore changes in the A13 region's connectome after 6-OHDA lesions. These alterations involved a complex rewiring of neural circuits, impacting both afferent and efferent projections. In summary, this study unveiled the pro-locomotor effects of A13 region photoactivation in both healthy and PD model mice. The study also indicates the preservation of A13 dopaminergic cells and the anatomical changes in neural circuitry following PD-like lesions that represent the anatomical substrate for a parallel motor pathway.

Strengths:

These findings hold significant relevance for the field of motor control, providing valuable insights into the organization of the motor system in mammals. Additionally, they offer potential avenues for addressing motor deficits in Parkinson's disease (PD). The study fills a crucial knowledge gap, underscoring its importance, and the results bolster its clinical relevance and overall strength.

The authors adeptly set the stage for their research by framing the central questions in the introduction, and they provide thoughtful interpretations of the data in the discussion section. The results section, while straightforward, effectively supports the study's primary conclusion-the pro-locomotor effects of A13 region stimulation, both in normal motor control and in the 6-OHDA model of brain damage.

Weaknesses:

(1) Anatomical investigation. I have a major concern regarding the anatomical investigation of plastic changes in the A13 connectome (Figures 4 and 5). While the methodology employed to assess the connectome is technically advanced and powerful, the results lack mechanistic insight at the cell or circuit level into the pro-locomotor effects of A13 region stimulation in both physiological and pathological conditions. This concern is exacerbated by a textual description of results that doesn't pinpoint precise brain areas or subareas but instead references large brain portions like the cortical plate, making it challenging to discern the implications for A13 stimulation. Lastly, the study is generally well-written with a smooth and straightforward style, but the connectome section presents challenges in readability and comprehension. The presentation of results, particularly the correlation matrices and correlation strength, doesn't facilitate biological understanding. It would be beneficial to explore specific pathways responsible for driving the locomotor effects of A13 stimulation, including examining the strength of connections to well-known locomotor-associated regions like the Pedunculopontine nucleus, Cuneiformis nucleus, LPGi, and others in the diencephalon, midbrain, pons, and medulla. Additionally, identifying the primary inputs to A13 associated with motor function would enhance the study's clarity and relevance.

The study raises intriguing questions about compensatory mechanisms in Parkinson's disease a new perspective with the preservation of dopaminergic cells in A13, despite the SNc degeneration, and the plastic changes to input/output matrices. To gain inspiration for a more straightforward reanalysis and discussion of the results, I recommend the authors refer to the paper titled "Specific populations of basal ganglia output neurons target distinct brain stem areas while collateralizing throughout the diencephalon from the David Kleinfeld laboratory." This could guide the authors in investigating motor pathways across different brain regions.

(2) Description of locomotor performance. Figure 3 provides valuable data on the locomotor effects of A13 region photoactivation in both control and 6-OHDA mice. However, a more detailed analysis of the changes in locomotion during stimulation would enhance our understanding of the pro-locomotor effects, especially in the context of 6-OHDA lesions. For example, it would be informative to explore whether the probability of locomotion changes during stimulation in the control and 6-OHDA groups. Investigating reaction time, speed, total distance, and even kinematic aspects during stimulation could reveal how A13 is influencing locomotion, particularly after 6-OHDA lesions. The laboratory of Whelan has a deep knowledge of locomotion and the neural circuits driving it so these features may be instructive to infer insights on the neural circuits driving movement. On the same line, examining features like the frequency or power of stimulation related to walking patterns may help elucidate whether A13 is engaging with the Mesencephalic Locomotor Region (MLR) to drive the pro-locomotor effects. These insights would provide a more comprehensive understanding of the mechanisms underlying A13-mediated locomotor changes in both healthy and pathological conditions.

(3) Figure 2 indeed presents valuable information regarding the effects of A13 region photoactivation. To enhance the comprehensiveness of this figure and gain a deeper understanding of the neurons driving the pro-locomotor effect of stimulation, it would be beneficial to include quantifications of various cell types:

• cFos-Positive Cells/TH-Positive Cells: it can help determine the impact of A13 stimulation on dopaminergic neurons and the associated pro-locomotor effect in healthy condition and especially in the context of Parkinson's disease (PD) modeling.

• cFos-Positive Cells /TH-Negative Cells: Investigating the number of TH-negative cells activated by stimulation is also important, as it may reveal non-dopaminergic neurons that play a role in locomotor responses. Identifying the location and characteristics of these TH-negative cells can provide insights into their functional significance.

Incorporating these quantifications into Figure 2 would enhance the figure's informativeness and provide a more comprehensive view of the neuronal populations involved in the locomotor effects of A13 stimulation.

(4) Referred to Figure 3. In the main text (page 5) when describing the animal with 6-OHDA the wrong panels are indicated. It is indicated in Figure 2A-E but it should be replaced with 3A-E. Please do that.

Summary of the Study after revision

The revised manuscript reflects significant efforts to improve clarity, organization, and data interpretation. The refinements in anatomical descriptions, behavioral analyses, and contextual framing have strengthened the manuscript considerably. However, the study still lacks direct causal evidence linking anatomical remodeling to behavioral improvements, and the small sample size in the anatomical analyses remains a concern. The authors have addressed many points raised in the initial review, but further acknowledgement of the exploratory nature of these findings would enhance the scientific rigor of the work.

Key Improvements in the Revision

The revised manuscript demonstrates considerable progress in clarifying data presentation, refining behavioral analyses, and improving the contextualization of anatomical findings. The restructuring of the anatomical section now provides greater precision in describing motor-related pathways, integrating terminology from the Allen Brain Atlas. The addition of new figures (Figures 4 and 5) strengthens the accessibility of these findings by illustrating key connectivity patterns more effectively. Furthermore, the correlation matrices have been adjusted to improve interpretability, ensuring that the presented data contribute meaningfully to the overall narrative of the study.

The authors have also made significant improvements in their behavioral analyses, particularly in the organization and presentation of locomotor data. Figure 3 has been revised to distinctly separate results from 6-OHDA and sham animals, providing a clearer comparison of locomotor outcomes. Additional metrics, such as reaction time, locomotion bouts, and movement speed, further enhance the granularity of the analysis, making the results more informative.

The discussion surrounding anatomical connectivity has also been strengthened. The revised manuscript now places greater emphasis on motor-related pathways and refines its analysis of A13 efferents and afferents. A newly introduced figure provides a concise summary of these connections, improving the contextualization of the anatomical data within the study's broader scope. Moreover, the authors have addressed the translational relevance of their findings by acknowledging the differences between optogenetic stimulation and deep brain stimulation (DBS). Their discussion now better situates the findings within existing literature on PD-related motor circuits, providing a more balanced perspective on the potential implications of A13 stimulation.

Remaining Concerns

Despite these substantial improvements, a number of critical concerns remain. The anatomical findings, though insightful, remain largely correlative and do not establish a causal link between structural remodeling and locomotor recovery. While the authors argue that these data will serve as a reference for future investigations, their necessity for the core conclusions of the study is not entirely clear. Additionally, while the anatomical data offer an interesting perspective on A13 connectivity, their direct relevance to the study's primary goal-demonstrating the role of A13 in locomotor recovery-remains uncertain. The authors emphasize that these data will be valuable for future research, yet their integration into the study's main narrative feels somewhat supplementary. Based on this last thought of the authors it is even more relevant another key limitation lying in the small sample size used for connectivity analyses. With only two sham and three 6-OHDA animals included, the statistical confidence in the findings is inherently limited. The absence of direct statistical comparisons between ipsilesional and contralesional projections further weakens the conclusions drawn from these anatomical studies. The authors have acknowledged that obtaining the necessary samples, acquiring the data, and analyzing them is a prolonged and resource-intensive process. While this may be a valid practical limitation, it does not justify the lack of a robust statistical approach. A more rigorous statistical framework should be employed to reinforce the findings, or alternative techniques should be considered to provide additional validation. Given these constraints, it remains unclear why the authors have not opted for standard immunohistochemistry, which could provide a complementary and more statistically accessible approach to validate the anatomical findings. Employing such an approach would not only increase the robustness of the results but also strengthen the study's impact by providing an independent confirmation of the observed structural changes.

---

## [Referee Report · Reviewer #2 (Public review)]

Summary:

The paper by Kim et al. investigates the potential of stimulating the dopaminergic A13 region to promote locomotor restoration in a Parkinson's mouse model. Using wild-type mice, 6-OHDA injection depletes dopaminergic neurons in the substantia nigra pars compacta, without impairing those of the A13 region and the ventral tegmentum area, as previously reported in humans. Moreover, photostimulation of presumably excitatory (CAMKIIa) neurons in the vicinity of the A13 region improves bradykinesia and akinetic symptoms after 6-OHDA injection. Whole-brain imaging with retrograde and anterograde tracers reveals that the A13 region undergoes substantial changes in the distribution of its afferents and projections after 6-OHDA injection, thus suggesting a remodeling of the A13 connectome. Whether this remodelling contributes to pro-locomotor effects of the photostimulation of the A13 region remains unknown as causality was not addressed.

Strengths:

Photostimulation of presumably excitatory (CAMKIIa) neurons in the vicinity of the A13 region promotes locomotion and locomotor recovery of wild-type mice 1 month after 6-OHDA injection in the medial forebrain bundle, thus identifying a new potential target for restoring motor functions in Parkinson's disease patients. The study also provides a description of the A13 region connectome pertaining to motor behaviors and how it changes after a dopaminergic lesion. Although there is no causal link between anatomical and behavioral data, it raises interesting questions for further studies.

Weaknesses:

Although CAMKIIa is a marker of presumably excitatory neurons and can be used as an alternative marker of dopaminergic neurons, some uncertainty remains regarding the phenotype of neurons underlying recovery of akinesia and improvement of bradykinesia.

Figure 4 is improved, but the results from the correlation analyses remain difficult to interpret, as they may reflect changes in various impaired brain regions independently of the A13 region. While the analysis offers a snapshot of correlated changes within the connectome, it does not identify which specific cell or axonal populations are actually increasing or decreasing. Although functional MRI connectome analyses are well-established, anatomical data seem less suitable for this purpose. How can one interpret correlated changes in anatomical inputs or outputs between two distinct regions?

Figure 5 is also improved, but there is room for further enhancement. As currently presented, it is difficult to distinguish the differences between the sham and 6-OHDA groups. The first column could compare afferents, while the second column could compare efferents. Given the small sample size, it would be more appropriate to present individual data rather than the mean and standard deviation.

Appraisal and impact

Although the behavioral experiments are convincing, the low number of animals in the anatomical studies is insufficient to make any relevant statistical conclusions due to extremely low statistical power.

---

## [Referee Report · Reviewer #3 (Public review)]

Kim, Lognon et al. present an important finding on pro-locomotor effects of optogenetic activation of the A13 region, which they identify as a dopamine-containing area of the medial zona incerta that undergoes profound remodeling in terms of afferent and efferent connectivity after administration of 6-OHDA to the MFB. The authors claim to address a model of PD-related gait dysfunction, a contentious problem that can be difficult to treat by dopaminergic medication or DBS in conventional targets. They make use of an impressive array of technologies to gain insight into the role of A13 remodeling in the 6-OHDA model of PD. The evidence provided is solid and the paper is well written, but there are several general issues that reduce the value of the paper in its current form, and a number of specific, more minor ones. Also some suggestions, that may improve the paper compared to its recent form, come to mind.

The most fundamental issue that needs to be addressed is the relation of the structural to the behavioral findings. It would be very interesting to see whether the structural heterogeneity in afferent/effects projections induced by 6-OHDA is related to the degree of symptom severity and motor improvement during A13 stimulation.

The authors provide extensive interrogation of large-scale changes in the organization of the A13 region afferent and efferent distributions. It remains unclear how many animals were included to produce Fig 4 and 5. Fig S5 suggests that only 3 animals were used, is that correct? Please provide details about the heterogeneity between animals. Please provide a table detailing how many animals were used for which experiment. Were the same animals used for several experiments?

While the authors provide evidence that photoactivation of the A13 is sufficient in driving locomotion in the OFT, this pro-locomotor effect seems to be independent of 6-OHDA induced pathophysiology. Only in the pole test do they find that there seems to be a difference between Sham vs 6-OHDA concerning effects of photoactivation of the A13. Because of these behavioral findings, optogenic activation of A13 may represent a gain of function rather than disease-specific rescue. This needs to be highlighted more explicitly in the title, abstract and conclusion.

The authors claim that A13 may be a possible target for DBS to treat gait dysfunction. However, the experimental evidence provided (imparticular lack of disease-specific changes in the OFT) seem insufficient to draw such conclusions. It needs to be highlighted that optogenetic activation does not necessarily have the same effects as DBS (see the recent review from Neumann et al. in Brain: https://pubmed.ncbi.nlm.nih.gov/37450573/). This is important because ZI-DBS so far had very mixed clinical effects. The authors should provide plausible reasons for these discrepancies. Is cell-specificity, that only optogenetic interventions can achieve, necessary? Can new forms of cyclic burst DBS achieve similar specificity (Spix et al, Science 2021)? Please comment.

In a recent study, Jeon et al (Topographic connectivity and cellular profiling reveal detailed input pathways and functionally distinct cell types in the subthalamic nucleus, 2022, Cell Reports) provided evidence on the topographically graded organization of STN afferents and McElvain et al. (Specific populations of basal ganglia output neurons target distinct brain stem areas while collateralizing throughout the diencephalon, 2021, Neuron) have shown similar topographical resolution for SNr efferents. Can a similar topographical organization of efferents and afferents be derived for the A13/ ZI in total?

In conclusion, this is an interesting study that can be improved taking into consideration the points mentioned above.

---

## [Author Response]

The following is the authors’ response to the previous reviews

**Reviewer #2 (Public review):**
Summary:The paper by Kim et al. investigates the potential of stimulating the dopaminergic A13 region to promote locomotor restoration in a Parkinson's mouse model. Using wild-type mice, 6-OHDA injection depletes dopaminergic neurons in the substantia nigra pars compacta, without impairing those of the A13 region and the ventral tegmentum area, as previously reported in humans. Moreover, photostimulation of presumably excitatory (CAMKIIa) neurons in the vicinity of the A13 region improves bradykinesia and akinetic symptoms after 6-OHDA injection. Whole-brain imaging with retrograde and anterograde tracers reveals that the A13 region undergoes substantial changes in the distribution of its afferents and projections after 6-OHDA injection, thus suggesting a remodeling of the A13 connectome. Whether this remodelling contributes to pro-locomotor effects of the photostimulation of the A13 region remains unknown as causality was not addressed.Strengths:Photostimulation of presumably excitatory (CAMKIIa) neurons in the vicinity of the A13 region promotes locomotion and locomotor recovery of wild-type mice 1 month after 6-OHDA injection in the medial forebrain bundle, thus identifying a new potential target for restoring motor functions in Parkinson's disease patients. The study also provides a description of the A13 region connectome pertaining to motor behaviors and how it changes after a dopaminergic lesion. Although there is no causal link between anatomical and behavioral data, it raises interesting questions for further studies.

Thank you for the comments.

Weaknesses:Although CAMKIIa is a marker of presumably excitatory neurons and can be used as an alternative marker of dopaminergic neurons, some uncertainty remains regarding the phenotype of neurons underlying recovery of akinesia and improvement of bradykinesia.

The primary objective was to focus on a population of neurons that could contribute to functional recovery, with a long-term translational focus in mind. We have followed up on this by creating a rat-based DBS model of stimulating the A13 region (Bisht et al 2025). We agree that the next steps are to genetically dissect the circuits, and we have made a start on this with our recent publication (Sharma et al 2024).

Figure 4 is improved, but the results from the correlation analyses remain difficult to interpret, as they may reflect changes in various impaired brain regions independently of the A13 region. While the analysis offers a snapshot of correlated changes within the connectome, it does not identify which specific cell or axonal populations are actually increasing or decreasing. Although functional MRI connectome analyses are well-established, anatomical data seem less suitable for this purpose. How can one interpret correlated changes in anatomical inputs or outputs between two distinct regions?

We appreciate the reviewer's thoughtful comment regarding the interpretability of the correlation analyses in Figure 4. We fully acknowledge that our anatomical data cannot establish causality or identify specific cell types or axonal populations undergoing changes following unilateral nigrostriatal degeneration. However, our intent with this analysis was not to infer mechanistic pathways but rather to provide a systems-level overview of how the global organization of A13 efferents and afferents is altered following 6-OHDA lesioning. By calculating proportions of total inputs and outputs and comparing them across brain regions, we aimed to control for variability in labeling and highlight relative shifts in network organization. The correlation matrices are intended to capture coordinated changes in input/output distribution patterns, effectively reflecting how groups of regions co-vary in their input to or output from the A13 region. In our case, we used correlation analysis to identify how input and output distributions across brain regions reorganize as a network following 6-OHDA lesioning. For example, a positive correlation between inputs from Region A and Region B to the A13 suggests that across animals, when input from Region A is relatively high, input from Region B tends to be high as well, indicating that connectivity from these regions to the A13 may be co-regulated or affected similarly by the lesion. Conversely, a shift from positive to negative correlation may signal a divergence in how regions contribute to the A13 connectome after nigrostriatal degeneration (e.g., increased connectivity to Region A compared to reduced connectivity to Region B). Thus, these patterns offer new insight into the broader reorganization of the A13 connectome and may serve as systems-level signatures of altered anatomical organization, providing a foundation for future mechanistic investigations using circuit-specific tools. We have revised the text to better emphasize the correlative and descriptive nature of these analyses and to clarify that they serve as a hypothesis-generating exploration. Future studies using cell type- and/or projection-specific functional manipulations will be essential to determine the causal roles of these reorganized circuits. We believe our use of this method is justified in the context of exploring broad, lesion-induced network reorganization, and we hope this additional context helps clarify the purpose and limitations of our approach.

Figure 5 is also improved, but there is room for further enhancement. As currently presented, it is difficult to distinguish the differences between the sham and 6-OHDA groups. The first column could compare afferents, while the second column could compare efferents. Given the small sample size, it would be more appropriate to present individual data rather than the mean and standard deviation.

We have reorganized Figure 5 as suggested.

Appraisal and impactAlthough the behavioral experiments are convincing, the low number of animals in the anatomical studies is insufficient to make any relevant statistical conclusions due to extremely low statistical power.

See previous comments on this.

**Reviewer #2 (Recommendations for the authors):**
Points that need to be addressed:Figure S1 is supposed to illustrate the percentage of expression in all mice, but the number of mice does not match (n=3 and 3 in Figure S1 versus n=5 and 6 in Figure 1). Revise the legend or add the missing data.

We have added the additional data to this graph (Figure 2 – figure supplement 1) and have separated out 6-OHDA and sham mice for clarity.

Page 4: "There was also an increase in the number of ChR2 cells with c-fos labeling in 6-OHDA ChR2 mice compared to the 6-OHDA eYFP mice. However, there was no net increase in TH+ cells labelled with ChR2 and c-Fos suggesting a heterogeneous population of activated cells." A quantification will be necessary to advance this conclusion.

We were able to determine that there was a trend of increased c-Fos intensity within the A13 region following photostimulation. However, the variability in the data makes it premature to comment on the TH co-localization and we have deleted this statement.

Figure 3: The choice of red and green could be a problem for color-blind people.

Thank you - switched to orange and cyan instead.

Page 7, 4th paragraph: "6-OHDA mice demonstrated significantly greater descent times than sham mice (Figure 3L, p<0.01)." This is not what is shown in the Figure 3L.

We made changes in the legend and text to clarify.

Page 7, last line: PT abbreviation should be introduced in parentheses at the beginning of this section.

Removed the abbreviation.

Figure S4A: The authors should show data for the VTA or refer to the quantification of Figure S4G in the text.

Now referenced correctly in the text.

Figure S7 and S8 are not referenced in the results or methods.

References added to text.

Double-check the formatting of some references: L.-X. Li et al, 2021, L. Kim et al., 2021.

References checked and corrected.